# Similar GABA$_A$ receptor subunit composition in somatic and axon initial segment synapses of hippocampal pyramidal cells

Katalin Kerti-Szigeti, Zoltan Nusser*

Laboratory of Cellular Neurophysiology, Institute of Experimental Medicine, Hungarian Academy of Sciences, Budapest, Hungary

**Abstract** Hippocampal pyramidal cells (PCs) express many GABA$_A$R subunit types and receive GABAergic inputs from distinct interneurons. Previous experiments revealed input-specific differences in $\alpha$1 and $\alpha$2 subunit densities in perisomatic synapses, suggesting distinct IPSC decay kinetics. However, IPSC decays evoked by axo-axonic, parvalbumin- or cholecystokinin-expressing basket cells were found to be similar. Using replica immunogold labeling, here we show that all CA1 PC somatic and AIS synapses contain the $\alpha$1, $\alpha$2, $\beta$1, $\beta$2, $\beta$3 and $\gamma$2 subunits. In CA3 PCs, 90% of the perisomatic synapses are immunopositive for the $\alpha$1 subunit and all synapses are positive for the remaining five subunits. Somatic synapses form unimodal distributions based on their immunoreactivity for these subunits. The $\alpha$2 subunit densities in somatic synapses facing Cav2.1 (i.e. parvalbumin) or Cav2.2 (cholecystokinin) positive presynaptic active zones are comparable. We conclude that perisomatic synapses made by three distinct interneuron types have similar GABA$_A$ receptor subunit content.

*For correspondence: nusser@ koki.hu

**Competing interests:** The authors declare that no competing interests exist.

## Introduction

A salient feature of cortical microcircuits is the presence of distinct types of GABA-releasing interneurons (INs; *Freund and Buzsáki, 1996*; *Somogyi and Klausberger, 2005*; *Ascoli et al., 2008*; *Krook-Magnuson et al., 2012*; *DeFelipe et al., 2013*). Decades of extensive research revealed dozens of different IN types based on their developmental origin, dendritic and axonal arborization, postsynaptic targets and gene expression profiles. Recent in vivo electrophysiological, imaging and optogenetic studies demonstrated that distinct IN types play unique roles in memory formation and retrieval, network oscillations and sensory perception (*Klausberger and Somogyi, 2008*; *Lovett-Barron et al., 2012*; *Varga et al., 2012*; *Kepecs and Fishell, 2014*). It is not surprising that a distal dendrite-innervating IN has a different role compared to one with axons synapsing on pyramidal cell (PC) somata. However, there are multiple types of INs that innervate the perisomatic region of PCs (axo-axonic cell (AAC), parvalbumin- (PV[+]) and cholecystokinin-expressing (CCK[+]) basket cells), yet participate in distinct network computations as inferred from their differential firing during oscillations (*Klausberger et al., 2003*, *2005*; *Varga et al., 2012*). For example, hippocampal PV[+] and CCK[+] basket cells receive similar excitatory inputs and provide their output to the same subcellular domains of PCs (somata and proximal dendrites), but fire in different phases of theta oscillations, and only PV[+] cells are active during sharp wave-associated ripples (*Klausberger et al., 2005*). To achieve this, they must utilize different molecular machinery for integrating their synaptic inputs, generating their action potential outputs and releasing GABA. An elegant example of their molecular specialization is that the release of GABA from the axon terminals of CCK[+] basket cells is

mediated by Cav2.2 voltage-gated Ca$^{2+}$ channels and is under the control of CB1 cannabinoid receptors. In contrast, the release of GABA from PV$^+$ basket cells is exclusively mediated by Cav2.1 channels and is regulated by m2 muscarinic and μ opiate receptors (reviewed by *Freund and Katona, 2007*). An even more intriguing molecular specialization was suggested based on high-resolution postembedding immunogold localization of GABA$_A$ receptors (GABA$_A$Rs). It was shown that PC somatic synapses facing PV$^+$ axon terminals contain mainly α1 subunit-containing GABA$_A$Rs (*Klausberger et al., 2002*) whereas somatic synapses opposite to CCK$^+$ boutons and AIS synapses predominantly contain GABA$_A$Rs incorporating the α2 subunit (*Nusser et al., 1996*; *Fritschy et al., 1998*; *Nyiri et al., 2001*). Such a presynaptic input-specific distribution of distinct postsynaptic receptor types held a great potential for the selective pharmacological manipulation of cortical networks in various psychiatric disorders (e.g. schizophrenia or anxiety; reviewed by *Lewis et al., 2005*; *Rudolph and Mohler, 2014*).

Synaptic GABA$_A$Rs with distinct subunit compositions confer different decay kinetics to the postsynaptic responses (*Verdoorn et al., 1990*; *Tia et al., 1996*; *Lavoie et al., 1997*; *Vicini, 1999*; *Bianchi et al., 2001*). The decay time of IPSCs mediated by α2 subunit-containing GABA$_A$Rs is 3–4-times slower compared to that of IPSCs mediated by α1 subunit-containing GABA$_A$Rs (*Eyre et al., 2012*). These results predict that AAC and CCK$^+$ basket cell-evoked IPSCs should have decay times several fold slower compared to the ones evoked by PV$^+$ basket cells. However, this is in contrast with the rather uniform decays of fast rising (likely perisomatic) spontaneous IPSCs recorded from PCs (*Nusser et al., 2001*). Furthermore, *Szabo et al. (2010)* provided direct evidence for the similar decays of unitary IPSCs evoked by AAC, PV$^+$ and CCK$^+$ basket cells in CA3 PCs. In addition, using genetically modified animals, *Heistek et al. (2013)* demonstrated that fast-spiking, PV$^+$ basket cells activate α2 subunit-containing GABA$_A$Rs in CA3 PCs. It is not only these functional studies that have indicated the lack of segregation of α1 and α2 subunit-dominant GABA$_A$Rs to distinct somatic synapses, as two recent studies using highly sensitive immunohistochemical approaches have also shown the presence of α1 and α2 subunits in all perisomatic synapses without any indication of multiple synapse populations (*Kasugai et al., 2010*; *Panzanelli et al., 2011*). However, from the presence of two unimodal distributions, a negative correlation between these two α subunits cannot be excluded; i.e. synapses that contain many α1 subunits could have few α2 subunits, and vice versa. Furthermore, almost nothing is known about the quantitative relationship between different β subunits in PC somatic and AIS synapses, although different β subunit isoforms were shown to be responsible for the polarized distribution of recombinant GABA$_A$Rs (*Connolly et al., 1996*). Furthermore, the β2 subunit has a critical role in a form of inhibitory synaptic plasticity observed at a cerebellar synapse (*He et al., 2015*), demonstrating its crucial involvement in setting the synaptic strength.

Using the highly sensitive, electron microscopy (EM) face-matched sodium dodecyl sulphate-digested freeze-fracture replica immunolabeling technique (SDS-FRL; *Hagiwara et al., 2005*), here we quantitatively analyzed the relative abundance of two α, three β and a γ subunit isoforms in hippocampal CA1 and CA3 PC AIS and somatic synapses to reveal the subunit composition of GABA$_A$Rs in three types of perisomatic region-targeting IN synapses.

## Results

### GABA$_A$Rs fracture to both faces of the plasma membrane

The basic principle of SDS-FRL is that following random fracturing of frozen tissue, the fracturing plane often crosses the lipid bilayer of plasma membranes with transmembrane proteins remaining either in the protoplasmic (P-face) or extracellular (E-face) half of the membrane (*Fujimoto, 1995*; *Rash et al., 1998*; *Masugi-Tokita and Shigemoto, 2007*). Carbon-platinum-carbon coating of the surfaces stabilizes the phospholipids and transmembrane proteins in the replica, allowing their visualization with specific antibodies. If an antibody recognizes intracellular epitope(s), specific immunolabeling will be detected in the P-face of the membrane, whereas extracellular epitopes will be visualized on the E-face. The most widely used application of SDS-FRL is to perform quantitative comparisons between distinct subcellular compartments. However, the most straightforward interpretation of the results (i.e. twice as many gold particles reflects twice as many proteins) is based on an assumption that has not been tested: If one compartment (e.g. synapse 'X') contains twice as many gold particles than another one (e.g. synapse 'Y'), the possibility that this is the consequence

of a larger fraction of the protein fracturing to the appropriate membrane, and not a difference in the density of the protein, has not been excluded. For example, two synapses could have an identical receptor density, but if 80% of the synaptic receptors fractured to the P-face in synapse 'X', but only 40% in synapse 'Y', SDS-FRL would result in twice as many gold particles in the P-face of synapse 'X', leading to an erroneous interpretation that synapse 'X' has more receptors. However, at the same time, an antibody that recognizes an epitope on the opposite side (E-face) would label the synapse 'Y' with 3-times as many gold particles as synapse 'X' (60% versus 20%), resulting in a negative correlation between the labels obtained in E- and P-faces.

As a prelude to our main experiments, we assessed the fracturing properties of the GABA$_A$R $\gamma$2 subunit in hippocampal CA1 PC somatic synapses using the face-matched mirror replica labeling method (*Figure 1A–D*). Somatic P-face membrane areas with high intramembrane particle (IMP) densities contained a large number of gold particles labeled with an anti-$\gamma$2$_{(Rb;\ aa319-366)}$ antibody. The complementary E-face membranes also contained many gold particles labeled with an anti-$\gamma$2 antibody recognizing an extracellular epitope ($\gamma$2$_{(Rb;\ aa39-67)}$). Gold particle numbers on both faces strongly and significantly correlated with the synaptic areas (*Figure 1E*), consistent with the results of many previous studies. Importantly, we found a strong positive correlation between P- and E-face gold particle numbers (*Figure 1F*), demonstrating that GABA$_A$Rs fracture to both faces of the plasma membrane and the synapse-to-synapse variability in gold particle number is not the

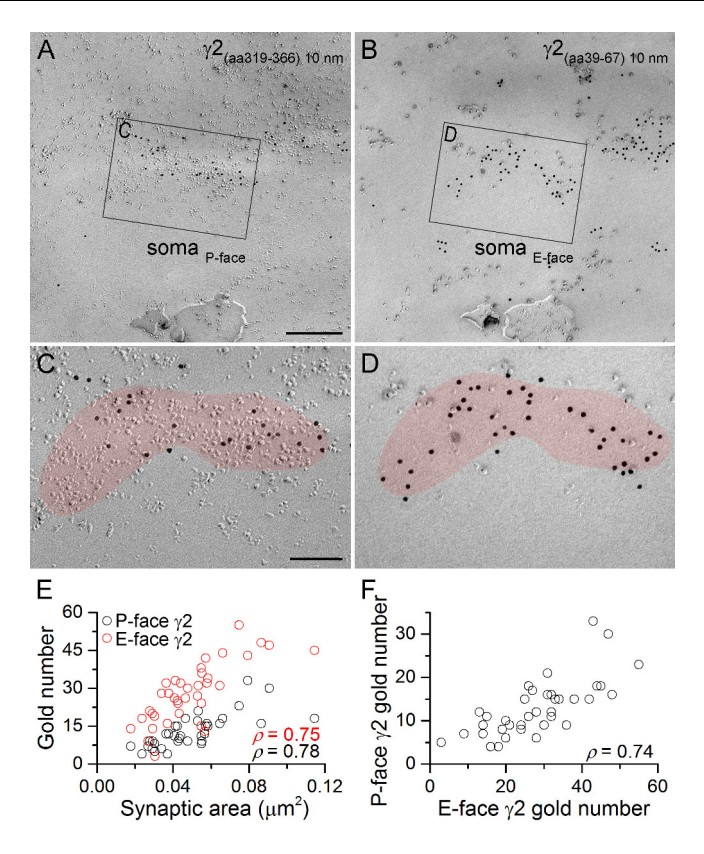

**Figure 1.** GABA$_A$ receptors fracture to both sides of the plasma membrane. (A, B) Face-matched mirror replica image of a CA1 pyramidal cell (PC) soma containing two GABAergic synapses identified on the P-face by an antibody directed against an intracellular epitope (A) and on the E-face by an antibody recognizing an extracellular epitope (B) of the $\gamma$2 subunit. (C, D) High magnification images of the boxed areas in (A) and (B). (E) The number of gold particles labeling the $\gamma$2 subunit scales with the synaptic area (Spearman correlation, $\rho = 0.78$ for the P-face $\gamma$2 and $\rho = 0.75$ for the E-face $\gamma$2, p<0.001 for both antibodies, n = 37 synapses, one rat). (F) The number of gold particles labeling the $\gamma$2 subunit on the P-face shows a strong positive correlation with the number of gold particles labeling the $\gamma$2 subunit on the E-face ($\rho$=0.74, p<0.001). Scale bars: (A, B) 250 nm; (C, D) 100 nm.

consequence of synapse-to-synapse variability in fracturing to E- and P-faces. Thus, for proteins where antibodies against both intra- and extracellular epitopes are not available (i.e. the vast majority of proteins), quantitative comparisons between different subcellular compartments can be reliably made with a single antibody.

## All GABAergic synapses in PC somata and AISs contain the γ2 subunit and neuroligin-2

First, we asked whether every GABAergic synapse on hippocampal PC soma and AIS contains the γ2 subunit and how much synapse-to-synapse variability there is in GABA$_A$R number and density. PC somata can be easily identified in replicas as large triangular cell bodies located in a band corresponding to the *stratum pyramidale*. In contrast, fractured plasma membranes of AISs cannot be recognized based on their position within the replicas or on morphological criteria, requiring their molecular identification. Thus, throughout the study, we used immunoreactions for Nav1.6 or Kv1.1 subunits and defined a membrane as an AIS if it was either Nav1.6 or Kv1.1 immunopositive (*Lorincz and Nusser, 2010*; *Kirizs et al., 2014*). To avoid complications arising from potential competition between different antibodies on the same replica, the mirror replica method was used for the co-localization of the γ2 subunit and neuroligin-2 (NL-2), a GABAergic synapse-specific cell adhesion molecule (*Figures 2* and *3*). Immunogold particles for NL-2 were clustered over small areas of somatic and AIS P-face membranes that contained a high density of IMPs, characteristic of GABAergic synapses (*Kasugai et al., 2010*). The complementary E-face membrane areas always contained many gold particles labeling for the γ2 subunit (*Figures 2A–D*, *3A–D*). When the γ2 subunit-immunoreacted replica was searched and somatic and AIS areas with strong γ2 subunit labeling were found, the corresponding areas on the opposite replica were always immunopositive for NL-2. The distributions of somatic and AIS synapses of both CA3 and CA1 PCs are unimodal based on their NL-2 and γ2 subunit content (*Figures 2H,I*, *3H,I*), indicating the lack of multiple synapse populations. The number of gold particles for both NL-2 and the γ2 subunit in AIS and somatic synapses shows a positive correlation with the synaptic area (*Figures 2E,F*, *3E,F*), demonstrating that the main source of variability in the postsynaptic receptor number is the variance in the synaptic area. There was large variability in the area of AIS and somatic synapses (CA3 AIS: CV = 0.6, n = 161; CA1 AIS: CV = 0.4, n = 164; CA3 soma: CV = 0.7, n = 1105; CA1 soma: CV = 0.7, n = 767). Statistical comparisons of the four compartments revealed that CA1 AIS synapses are significantly smaller than CA1 somatic and CA3 AIS synapses (*Figure 3L*). Finally, we found positive correlations between the number of gold particles for the γ2 subunit and NL-2 in individual AIS and somatic synapses (*Figures 2G*, *3G*). Comparisons of NL-2 densities between CA3 and CA1 PC AIS and somatic synapses showed no significant differences (for normalization see Materials and methods; *Figure 3J*). In contrast, we found that normalized γ2 subunit densities were significantly higher in CA3 and CA1 AIS synapses than in somatic synapses of both regions (*Figure 3K*). These results demonstrate that all GABAergic synapses on the soma and AIS of both CA1 and CA3 PCs contain the γ2 subunit and NL-2, enabling their use as GABAergic synapse-specific markers.

## Somatic and AIS synapses form unimodal distributions based on their α1 and α2 subunit content

Next, we asked whether the two most abundant α subunits expressed by hippocampal PCs, the α1 and α2 subunits, show input-specific distributions. First, we carried out face-matched mirror replica labeling for NL-2 and the α1 subunit (*Figure 4*). Replicas immunoreacted for Nav1.6 and NL-2 were first screened and when a NL-2-positive GABAergic synapse was found on P-face somatic or AIS membranes, its corresponding E-face membrane was photographed and the α1 subunit content was quantitatively analyzed (*Figure 4A–D*). This sampling method resulted in an unbiased synapse population and revealed that all somatic and AIS synapses are immunopositive for the α1 subunit on CA1 PCs (*Figure 4I*), but that ~10% of CA3 PC synapses are immunonegative for the α1 subunit (AIS: 4 out of 50; soma: 44 out of 402; *Figure 4H*). The density of the α1 subunit was twice as high in CA1 compared to CA3 perisomatic synapses (*Figure 4G*). Immunonegative synapses on CA3 PCs either lack the α1 subunit or contain it at a low density that falls below the detection threshold of our method. However, the fact that >99% of synapses in CA1 PCs are immunopositive, where the α1 density is higher, suggests that the latter possibility might be more likely.

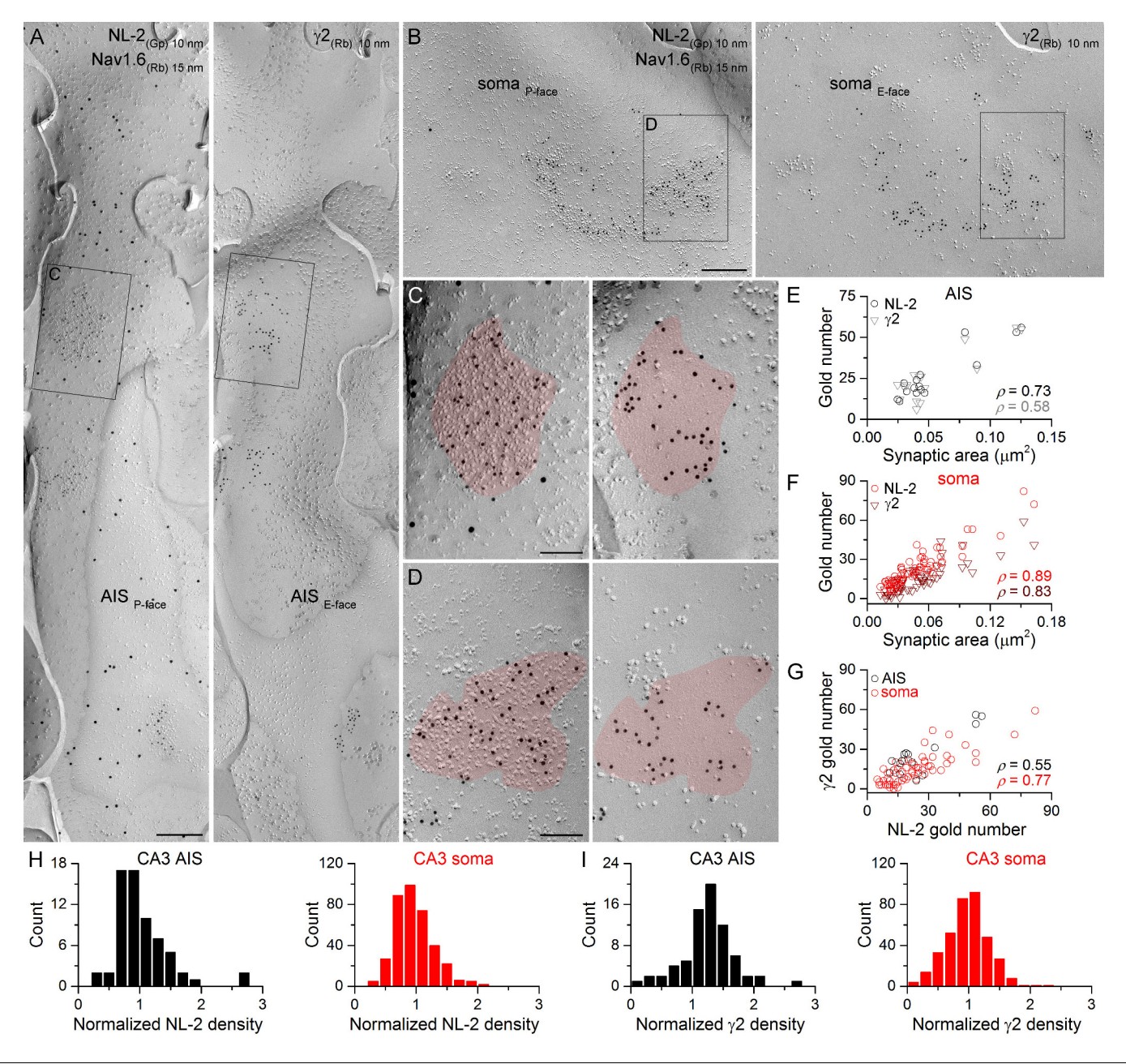

**Figure 2.** Identification of axon initial segment (AIS) and somatic GABAergic synapses in CA3 PCs using replica immunolabeling. (**A**) Mirror replica image of an AIS identified by immungold labeling for the Nav1.6 subunit (15 nm gold) on the P-face. Immunogold particles labeling neuroligin-2 (NL-2; 10 nm gold) are accumulated over dense intramembrane particle clusters characteristic for GABAergic synapses on the P-face of the AIS. The complementary E-face of the same membrane fragment contains clusters of gold particles labeling the γ2 subunit (10 nm gold). (**B**) A face-matched replica pair showing a fragment of a somatic plasma membrane containing two GABAergic synapses labeled for NL-2 on the P-face and the γ2 subunit on the E-face. (**C, D**) High magnification views of the boxed areas in (**A**) and (**B**). (**E, F**) The number of gold particles labeling for NL-2 and the γ2 subunit shows positive correlation with the synaptic area of AIS (**E**; Spearman correlation, ρ = 0.73, p=0.002 for NL-2 and ρ = 0.58,p=0.024 for γ2, n = 15 synapses, one rat) and somatic (**F**; ρ = 0.89 for NL-2 and ρ = 0.83 for γ2, p<0.001, n = 65 synapses, one rat) synapses of CA3 PCs. (**G**) Correlation between immunogold labeling for NL-2 and the γ2 subunit in AIS and somatic synapses (AIS: ρ = 0.55, p=0.035; soma: ρ=0.77, p<0.001, one rat). (**H, I**) Distributions of AIS (black) and somatic (red) synapses based on their normalized NL-2 (**H**; n = 65 for AIS and n = 369 for somatic synapses, data from five rats) and γ2 subunit (**I**; n = 72 for AIS and n = 367 for somatic synapses, data from three rats) densities. Scale bars: (**A, B**) 250 nm; (**C, D**) 100 nm.

The following source data is available for figure 2:

**Source data 1.** Data containing normalized NL-2 and γ2 subunit densities in CA3 AIS and somatic synapses are shown.

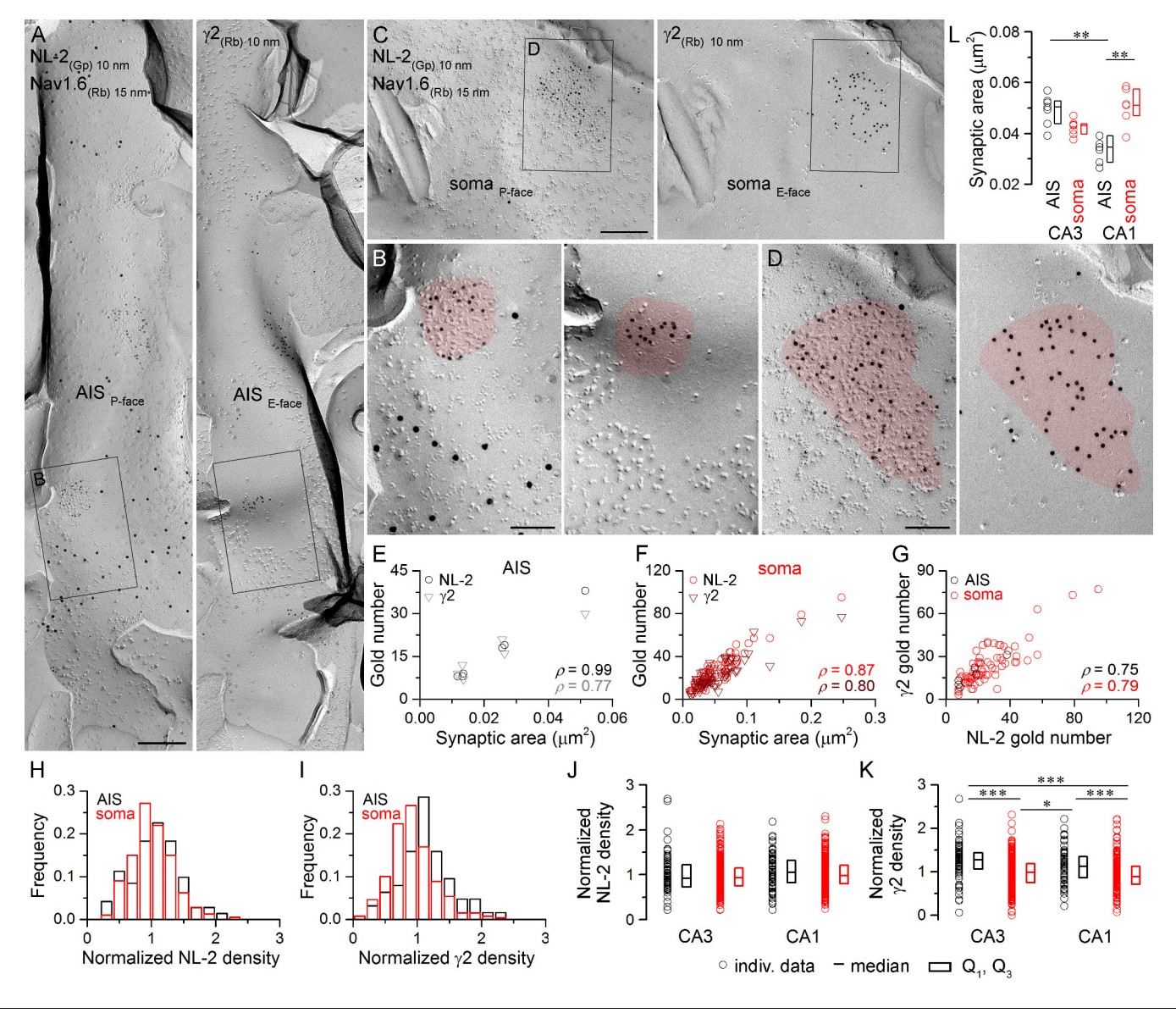

**Figure 3.** Identification of AIS and somatic GABAergic synapses in CA1 PCs with SDS-FRL. (**A**) Mirror replica image of an AIS double-labeled for Nav1.6 (15 nm gold) and NL-2 (10 nm gold) on the P-face, and the γ2 subunit (10 nm gold) on the complementary E-face of the same fragment of the plasma membrane. (**B**) High magnification view of the boxed areas shown in (**A**). (**C**) Mirror replica images showing a fragment of a somatic plasma membrane containing a GABAergic synapse labeled for NL-2 (10 nm gold) on the P-face and the γ2 subunit on the E-face (10 nm gold). (**D**) High magnification views of the boxed areas in (**C**). (**E, F**) The number of gold particles labeling for NL-2 and the γ2 subunit scales with synaptic area of AIS (**E**; Spearman correlation, ρ = 0.99, p<0.001 for NL-2 and ρ = 0.77, p=0.072 for γ2, n = 6 synapses, one rat) and somatic (**F**; ρ = 0.87 for NL-2 and ρ = 0.80 for γ2, p<0.001, n = 74 synapses, one rat) synapses of CA1 PCs. (**G**) Correlation between gold particle number for NL-2 and the γ2 subunit in AIS and somatic synapses (ρ=0.75, p=0.084 for AIS synapses and ρ = 0.79, p<0.001 for somatic synapses, respectively, one rat). (**H, I**) Distributions of normalized NL-2 (**H**; n = 71 AIS and n = 387 somatic synapses, data pooled from five rats) and γ2 subunit densities (**I**; n = 63 AIS and n = 259 somatic synapses, data pooled from three rats) in AIS (black) and somatic synapses (red). (**J**) Normalized density of NL-2 was not significantly different in AIS and somatic synapses in the CA3 and CA1 regions (Kruskal-Wallis test, p=0.064; pooled individual synaptic density values from five rats). (**K**) Normalized density of the γ2 subunit was significantly higher in AIS compared to somatic synapses in both CA3 and CA1 regions (Kruskal-Wallis test, p<0.001; Multiple comparisons of mean ranks, p<0.001 for both CA3 and CA1 regions; pooled individual synaptic density values from three rats). In addition, the γ2 subunit density was significantly higher in CA3 and CA1 AIS synapses compared to CA1 and CA3 somatic synapses (p<0.001 for CA3 AIS vs CA1 soma and p=0.029 for CA1 AIS vs CA3 soma). Plots show individual data (circles), medians (horizontal line) and lower and upper quartiles (Q1, Q3; boxes). (**L**) The area of AIS synapses was significantly smaller in the CA1 compared to the CA3 region (Kruskal-Wallis test, p<0.001; Multiple comparisons of mean ranks, p=0.005; n = 7 rats), and somatic synapses were significantly larger than AIS synapses in the CA1 region (p= 0.002). Open circles denote median synaptic areas

*Figure 3 continued on next page*

*Figure 3 continued*

from each rat, and the medians (horizontal line) and lower and upper quartiles (Q1, Q3; boxes) of animals are shown. Scale bars: (**A, C**) 250 nm; (**B, D**) 100 nm.

The following source data is available for figure 3:

**Source data 1.** Data containing normalized NL-2 and γ2 subunit densities in CA1 AIS and somatic synapses, and summary data for synaptic area measurements are shown.

Next we assessed the α2 subunit content of synapses (*Figure 5*) and found a significantly lower density in CA1 somatic synapses compared to the other three synapse populations (*Figure 5G*). Furthermore, the α2 subunit density was significantly lower in CA3 somatic synapses than in CA1 AIS synapses (*Figure 5G*). Despite these quantitative differences in the overall densities, no immunonegative synapse was found in any of these compartments. Somatic and AIS synapses on CA3 and CA1 PCs formed unimodal distributions (*Figure 5H,I*), indicating the lack of multiple synapse populations within these compartments. The similar variability in α2 subunit densities in AIS and somatic synapses (CA3 AIS: CV = 0.3, n = 46, CA3 soma: CV = 0.2, n = 470; CA1 AIS: CV = 0.2, n = 35, CA1 soma: CV = 0.3, n = 222) also suggests the lack of multiple somatic synapse populations based on their GABA$_A$R α2 subunit content.

## Co-localization of α1 and α2 subunits in individual synapses using the face-matched mirror replica technique

The lack of multiple somatic synapse populations based on their α1 and α2 subunit densities does not exclude the possibility that strongly α1 positive synapses are weakly α2 positive and vice versa. To examine this directly, we evaluated the α1 vs. α2 density ratio (α1:α2) in somatic and AIS synapses following double-labeling in mirror replicas (*Figure 6*). As demonstrated above all AIS and somatic GABAergic synapses contain the α2 subunit. Therefore labeling for the α2 subunit was used to identify GABAergic synapses on the P-face, while the mirror half of the synapse was photographed and its α1 subunit content was analyzed (*Figure 6A–C*). We found that strongly α2 subunit labeled synapses contain many immunogold particles for the α1 subunit (*Figure 6B*) and fewer gold particles are found for the α1 subunit if the α2 subunit labeling is weaker (*Figure 6C*). Comparison of the somatic α1 and α2 subunit densities showed a lack of negative correlation in both the CA1 and CA3 areas (*Figure 6D*).

It is important to note that in these reactions we were unable to identify somatic synapses established by PV[+] or CCK[+] presynaptic axon terminals, and therefore, first, we aimed at providing an indirect analysis of the somatic synapses. We compared the variability in the α1:α2 between AIS and somatic synapses. The AIS receives GABAergic inputs from a single IN type, whereas PC somata are innervated by two distinct types of basket cells; thus if the α1:α2 was different in PV[+] and CCK[+] synapses, a larger variability is predicted for the somatic synapses. However, our analysis revealed a lack of significant difference between the α1:α2 in AIS and somatic synapses in the CA1 (p=0.347, Kruskal-Wallis test, followed by multiple comparisons of mean ranks) and CA3 (p=0.053) regions (*Figure 6E*).

## Somatic synapses established by PV[+] and CCK[+] axon terminals have similar α2 subunit densities

To directly compare the α2 subunit content of somatic synapses established by PV[+] or CCK[+] axon terminals, we performed quadruple-labeling on face-matched replicas (*Figure 7*). One face of the replica was labeled for the α2 subunit (P-face) and the complementary replica was triple-labeled for the γ2 subunit (E-face) to identify GABAergic synapses, and for Cav2.1 (P-face) and Cav2.2 (P-face) subunits of the voltage-gated Ca$^{2+}$ channels to label the presynaptic AZs of PV[+] and CCK[+] axon terminals, respectively (*Lenkey et al., 2015*). Small randomly-fractured P-face membrane fragments of axon terminals are often attached to large E-face somatic plasma membranes. We selectively searched for such axon membranes that contained a partially fractured Cav2.1- or Cav2.2-immunoreactive AZ (*Figure 7A–C*). Sometimes, the fracturing plane traversed from the pre- to the postsynaptic plasma membrane within the synapse, resulting in Ca$^{2+}$ channel-labeled AZ membrane opposed

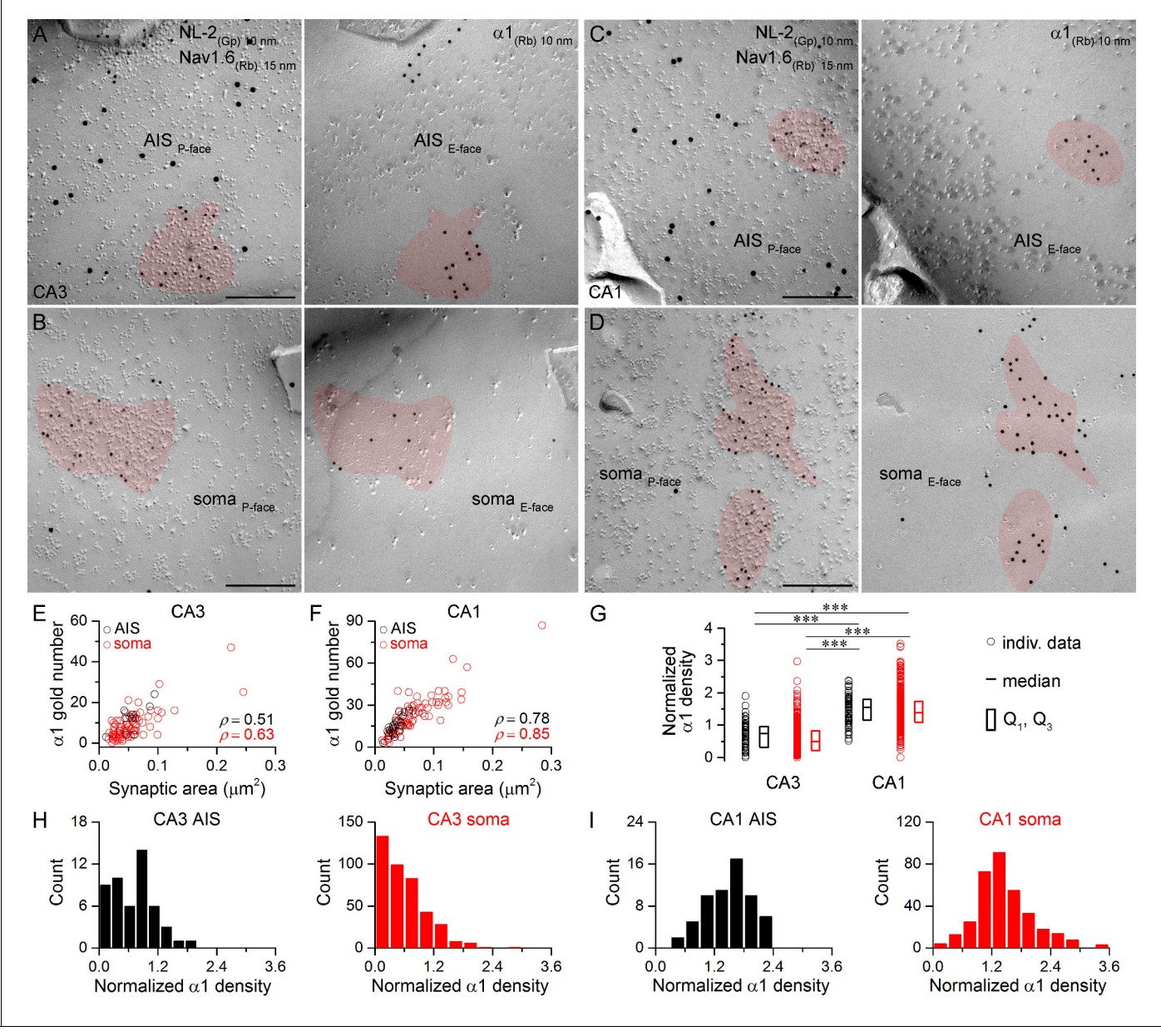

**Figure 4.** Hippocampal AIS and somatic synapses form unimodal distributions based on their α1 subunit densities. (**A–D**) High magnification images of double-replica pairs showing clusters of gold particles for the α1 subunit (10 nm gold) on the E-face of AIS (**A, C**) and somatic (**B, D**) synapses, which were identified on the complementary P-face by immunoreactivity for NL-2 (10 nm gold). The AIS was identified by immunolabeling for the Nav1.6 subunit (15 nm gold) on the P-face. (**E, F**) Immunogold particle labeling for the α1 subunit scales with the synaptic area in AIS and somatic synapses of CA3 (E; Spearman correlation, AIS: ρ = 0.51, p=0.112, n = 11; soma: ρ = 0.63, p<0.001, n = 84, one rat) and CA1 (**F**; AIS: ρ = 0.78, p<0.001, n = 23; soma: ρ = 0.85, p<0.001, n = 76, one rat) PCs. (**G**) The α1 subunit densities are significantly lower in CA3 AIS and somatic synapses compared to those in the CA1 region (Kruskal-Wallis test, p<0.001; Multiple comparisons of mean ranks, p<0.001 for both AIS and somatic synapses; pooled individual synaptic density values from four rats). Plots show individual data (circles), medians (horizontal line) and lower and upper quartiles (Q1, Q3; boxes). (**H, I**) Distributions of AIS and somatic synapses in the CA3 (**H**; AIS: n = 50; soma: n = 402 synapses) and CA1 regions (**I**; AIS: n = 61; soma: n = 337 synapses) based on their normalized α1 subunit densities. Scale bars: (**A–D**) 200 nm.

The following source data is available for figure 4:

**Source data 1.** Data containing normalized α1 subunit densities in CA3 and CA1 AIS and somatic synapses are shown.

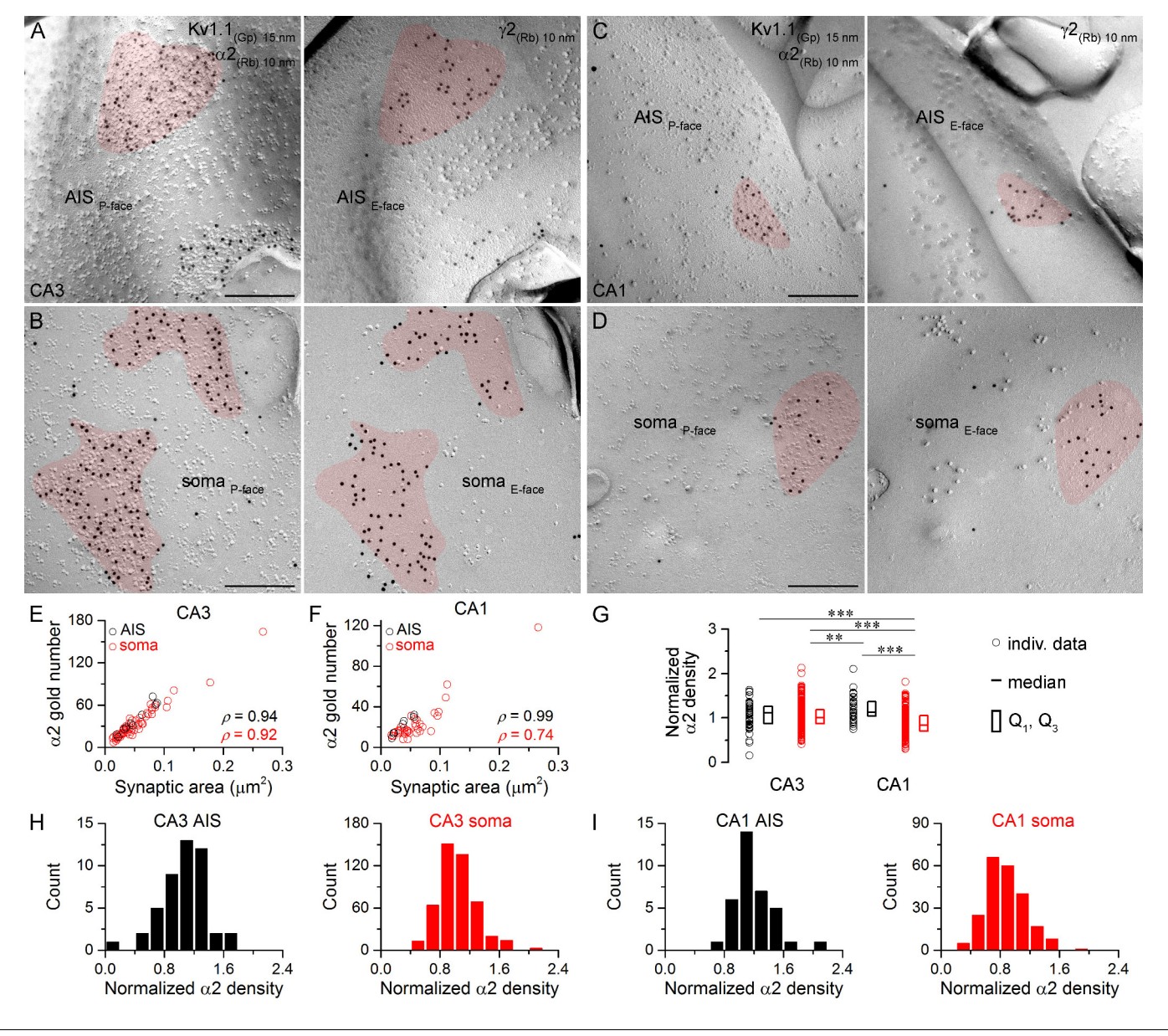

**Figure 5.** Hippocampal AIS and somatic synapses form unimodal distributions based on their α2 subunit densities. (A–D) High magnification images of double-replica pairs show the clustering of gold particles for the α2 subunit (10 nm gold) in AIS (A, C) and somatic (B, D) synapses on the P-face, identified on the complementary E-faces by the presence of gold particles for the γ2 subunit (10 nm gold). (E, F) The number of gold particles labeling the α2 subunit shows tight positive correlation with the synaptic area in AIS and somata of CA3 (E; Spearman correlation, AIS: ρ = 0.94, n = 10; soma: ρ = 0.92, n = 60; p<0.001; one rat) and CA1 PCs (F; AIS: ρ = 0.99, n = 7; soma: ρ = 0.74, n = 36; p<0.001; one rat). (G) Normalized α2 subunit densities are significantly lower in CA1 somatic synapses compared to CA1 AIS, CA3 somatic and CA3 AIS synapses (Kruskal-Wallis test, p<0.001; Multiple comparisons of mean ranks p<0.001 for all compartments; pooled individual synaptic density values from four rats). Furthermore, the α2 subunit density is significantly lower in CA3 somatic synapses than in CA1 AIS synapses (p<0.01). Plots show individual data (circles), medians (horizontal line) and lower and upper quartiles (Q1, Q3; boxes). (H, I) Distributions of AIS and somatic synapses in the CA3 (H; AIS: n = 46, soma: n = 470 synapses) and CA1 regions (I; AIS: n = 35, soma: n = 222 synapses) according to their normalized α2 subunit densities. Scale bars: (A–D) 200 nm.

The following source data is available for figure 5:

**Source data 1.** Data containing normalized α2 subunit densities in CA3 and CA1 AIS and somatic synapses are shown.

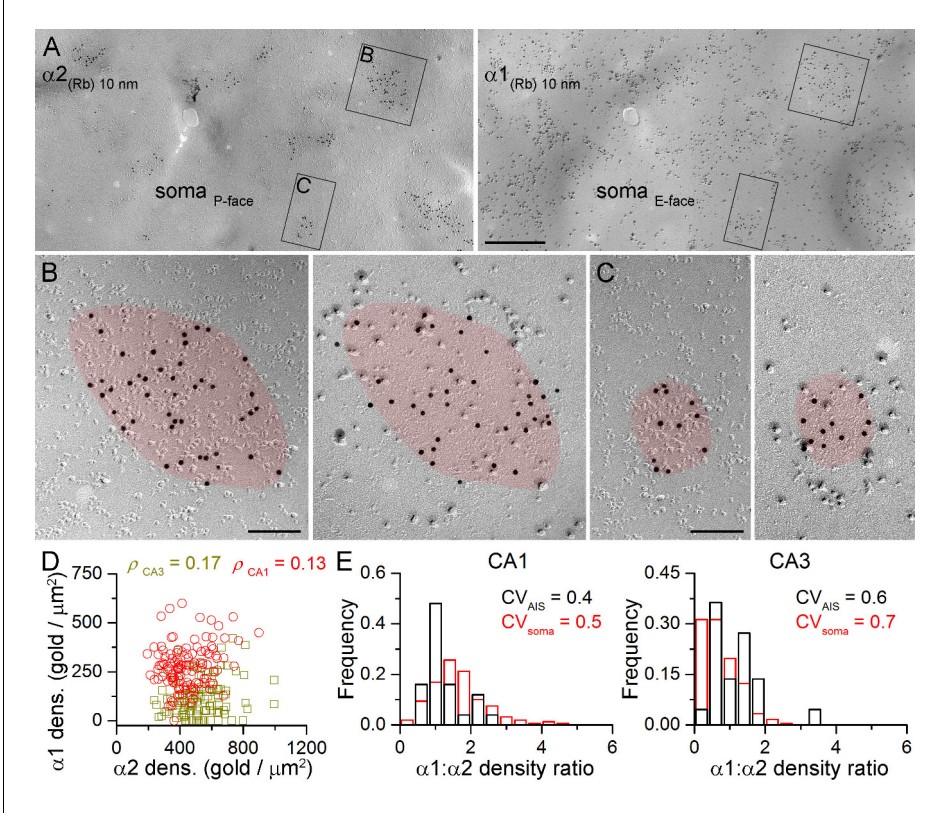

**Figure 6.** Face-matched mirror replica labeling for the α1 and α2 subunits in PC somatic synapses. (A) Low-magnification images of a double-replica pair depict a fragment of a CA1 PC soma, containing many GABAergic synapses labeled for the α2 subunit (10 nm gold) on the P-face and for the α1 subunit (10 nm gold) on the complementary E-face. (B, C) Two synapses illustrated in (A) are shown at a higher magnification. (D) No significant correlation was found between the α1 and α2 subunit densities of individual CA1 (red circles; Spearman correlation, ρ = 0.13, p=0.172, n = 119 synapses, one rat) and CA3 (olive squares; ρ = 0.17, p=0.075, n = 117 synapses, one rat) PC somatic synapses. (E) Distributions of α1:α2 density ratios in AIS and somatic synapses are shown for CA1 and CA3 PCs. Note the similar variability in the density ratio between somatic (red) and AIS (black) synapses (AIS: CV = 0.4, n = 25; soma: CV = 0.5; n = 160 in the CA1 region, and AIS: CV = 0.6, n = 22; soma: 0.7; n = 243 in the CA3 region; data pooled from two rats). Scale bars: (A) 500 nm; (B, C) 100 nm.

The following source data is available for figure 6:

**Source data 1.** Summary data for α1:α2 density ratio measurements are shown.

to its γ2 subunit-labeled postsynaptic partner. Once such a 'partially fractured' postsynaptic membrane was found, we photographed its complementary P-face on the mirror replica and quantified its α2 subunit content. In addition, we imaged all surrounding α2 subunit-immunoreactive synapses with unknown presynaptic elements on the same somatic plasma membrane (*Figure 7D*).

The number of α2 immunogold particles scaled with the size of the 'partially fractured' synapses for both Cav2.1[+] and Cav2.2[+] AZ-associated synapses (data not sown), similar to that found for complete synapses (*Figure 5E,F*). The densities of the α2 subunit were estimated from 'partial' postsynaptic membrane fragments attached to Cav2.1[+] and Cav2.2[+] AZs obtained from the CA1 (number of 'partial' synapses and total synaptic areas: Cav2.1: 35 and 0.8 $\mu m^2$; Cav2.2: 65 and 2.8 $\mu m^2$) and CA3 areas (Cav2.1: 54 and 1.3 $\mu m^2$; Cav2.2: 41 and 1.1 $\mu m^2$) of two rats. These experiments revealed similar α2 subunit densities in Cav2.1[+] and Cav2.2[+] AZ-associated postsynaptic membranes in both hippocampal regions (*Figure 7E,F*). In the CA1 area, these densities were not significantly different from the density obtained in the surrounding completely fractured synapses with unknown presynaptic elements (*Figure 7E*), but in the CA3 area the density of α2 subunit in the Cav2.1[+] AZ-

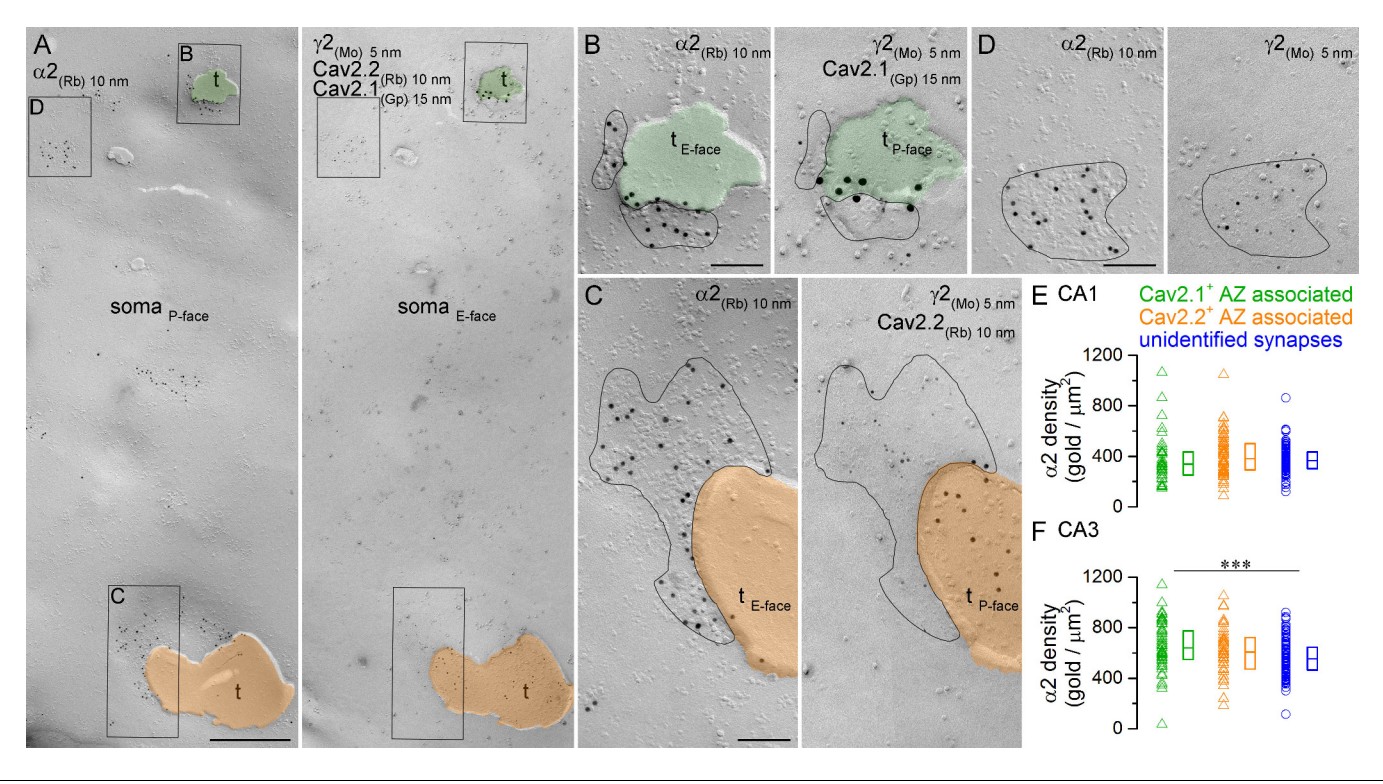

**Figure 7.** Comparison of immunogold labeling for the α2 subunit in synapses established by PV[+] or CCK[+] axon terminals on hippocampal PC somata. (A) Mirror replica images of a fragment of a PC somatic plasma membrane are shown at a low magnification. The P-face is immunolabeled for the α2 subunit (10 nm gold). On the E-face of the same plasma membrane, immunogold labeling for Cav2.1 (15 nm gold) and Cav2.2 (10 nm gold) was used to identify putative PV[+] (green overlay) and CCK[+] axon terminals (orange overlay), respectively. Additionally, immunolabeling for the γ2 subunit (5 nm) was used to visualize GABAergic synapses. Axon terminals (t) were identified as P-face membrane fragments attached to the E-face of the somatic plasma membrane. (B) High magnification view of the synapse established by a Cav2.1[+] axon terminal shown in (A). (C) High magnification image of the synapse established by a Cav2.2[+] axon terminal. (D) High magnification mirror replica image of a complete postsynaptic density of an unidentified synapse also shown in (A). (E) α2 subunit densities are similar in Cav2.1[+] and Cav2.2[+] active zone-associated synapses, and in unidentified somatic synapses of CA1 PCs (Kruskal-Wallis test, p=0.252; Cav2.1[+]: n = 35; Cav2.2[+]: n = 65; unidentified synapses: n = 103; data pooled from two rats). (F) Gold particle densities for the α2 subunit in Cav2.1[+] and Cav2.2[+] active zone-associated synapses, and in unidentified somatic synapses of CA3 PCs (Kruskal-Wallis test, p=0.002; Multiple comparisons of mean ranks between Cav2.1[+] and Cav2.2[+] synapses, p=0.579; Cav2.1[+] and unidentified synapses, p=0.001; Cav2.2[+] and unidentified synapses p=0.278; Cav2.1[+]: n = 54; Cav2.2[+]: n = 41; unidentified synapses: n = 129; data pooled from two rats). Individual data derived from partial synapse quantifications are shown as open triangles, and open circles denote data from complete synapse measurements. Medians (horizontal bars) with lower and upper quartiles (boxes) are shown for α2 densities in the three synapse type. Scale bars: (A) 500 nm; (B–D) 100 nm.

associated synapses was significantly higher than that in unidentified synapses (*Figure 7F*). The variability in α2 subunit density was also similar between the three synapse populations (CA1 area: Cav2.1[+]: CV = 0.5; Cav2.2[+]: CV = 0.4; unidentified: CV = 0.3; CA3 area: Cav2.1[+]: CV = 0.3; Cav2.2[+]: CV = 0.3; unidentified: CV = 0.2).

## Immunogold labeling for GABA$_A$R β1, β2 and β3 subunits in AIS and somatic synapses

Different α subunits have preferred β subunit partners in the pentameric GABA$_A$R complex. If distinct perisomatic synapses have GABA$_A$Rs with different subunit compositions, it is likely to be reflected in differential distributions of the β subunit isoforms as well. We obtained antibodies that provided specific immunolabeling against all β subunit isoform (β1, β2, β3) and co-localized them with the γ2 subunit in face-matched mirror replicas. Strong clustering of immunogold particles labeling the β1 (*Figure 8*), β2 (*Figure 9*) or β3 (*Figure 10*) subunits was found over IMP clusters characteristic of postsynaptic membranes on the P-face in both PC AISs and somata. Immunogold labeling for

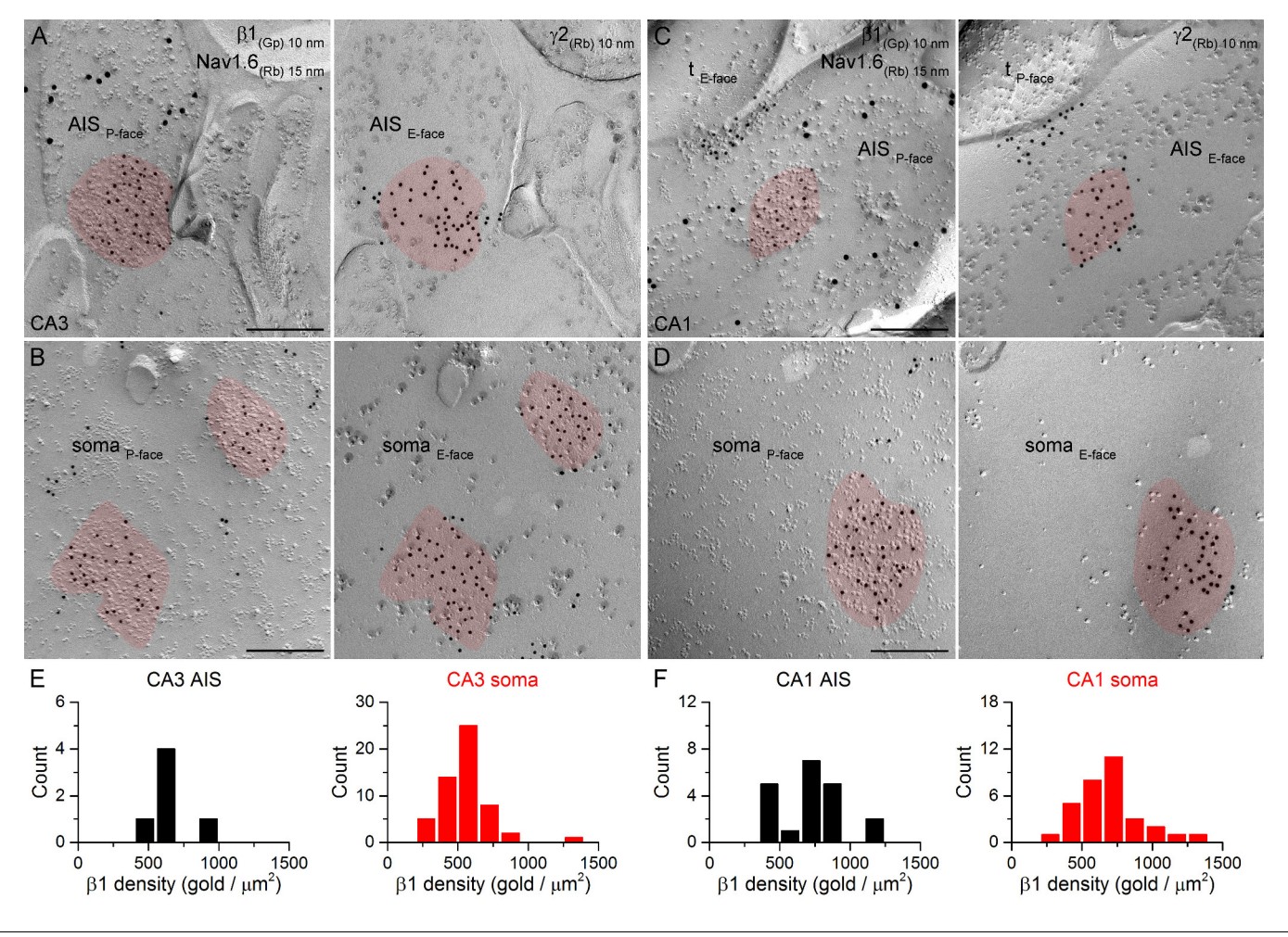

**Figure 8.** SDS-FRL labeling for the β1 subunit in hippocampal PC AIS and somatic synapses. (A–D) High magnification images of double-replica pairs show the accumulation of gold particles labeling the β1 subunit (10 nm gold) in the P-face of AIS (A, C) and somatic (B, D) synapses, which were identified on the complementary E-face by the presence of gold particles labeling for the γ2 subunit (10 nm gold). The AIS was identified by immunolabeling for the Nav1.6 subunit (15 nm gold). Note an axon terminal (t$_{E-face}$) facing a partial synapse immunolabeled for the β1 subunit of an AIS in the CA1 area (C). (E, F) Distributions of AIS (black) and somatic synapses (red) in the CA3 (E; AIS: n = 6, soma: n = 55, one rat) and CA1 (F; AIS: n = 20; soma: n = 32) regions based on their β1 subunit densities. Scale bars: (A–D) 200 nm.

the β subunits on the P-face always coincided with γ2 subunit labeling on the complementary E-face membrane and vice versa. Quantitative analysis of the reactions revealed that all AIS and somatic synapses in the CA3 and CA1 regions contained the β1, β2, and β3 subunits and that somatic synapses formed unimodal distributions based on their β subunit densities (*Figures 8E,F; 9E,F; 10E,F*).

## Immunofluorescent labeling for GABA$_A$Rs in AIS and somatic synapses

To inquire whether the results obtained with EM SDS-FRL are similar to those of light microscopy (LM) fluorescent localization of GABA$_A$Rs, we carried out double-immunofluorescent labeling for NL-2 and six GABA$_A$R subunits in the hippocampus of rats perfusion-fixed with a low-pH-based fixative (*Lorincz and Nusser, 2010*). Tissue prepared from very mildly-fixed rat brains showed robust and punctate immunolabeling for both NL-2 and the γ2 subunit in the hippocampus (*Figure 11* illustrates reactions in the CA3 region). Strongly immunoreactive puncta surrounded PC somata and ankyrin-G-labeled AISs. Qualitative evaluation of our reactions revealed that apparently all clusters were immunopositive for both NL-2 and the γ2 subunit, consistent with our SDS-FRL results. AISs were strongly and dominantly demarcated by immunoreactive puncta for both of these molecules, allowing their

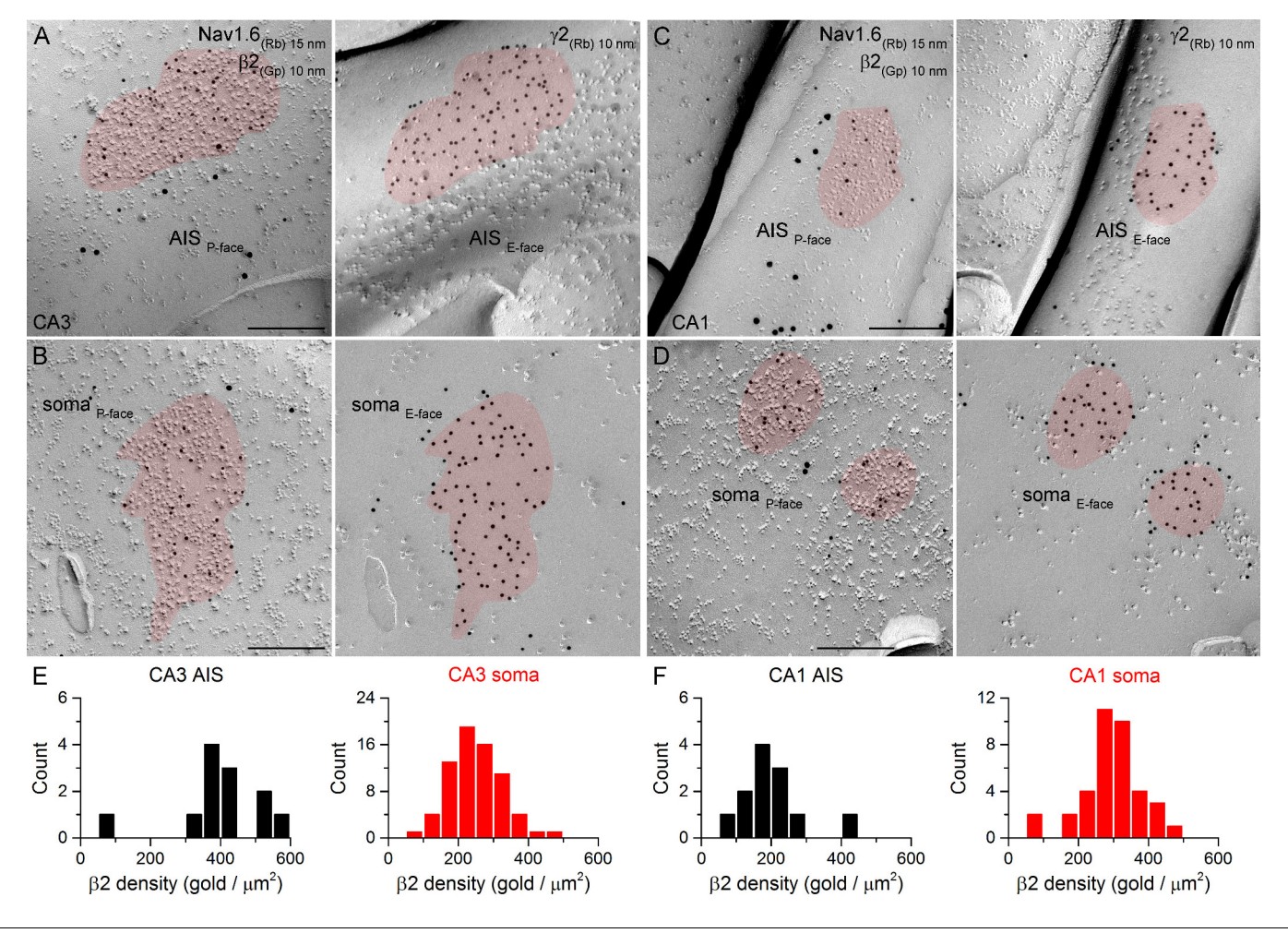

**Figure 9.** SDS-FRL labeling for the β2 subunit in hippocampal PC AIS and somatic synapses. (A–D) High magnification views of mirror replica images show the clustering of β2 subunit (10 nm gold) in AIS (A, C) and somatic (B, D) synapses on the P-face, identified on the complementary E-face by the presence of gold particles labeling for the γ2 subunit (10 nm gold). The AIS was identified by immunolabeling for the Nav1.6 subunit (15 nm gold). (E, F) Distributions of AIS (black) and somatic synapses (red) in the CA3 (E; AIS: n = 12; soma: n = 70, one rat) and CA1 (F; AIS: n = 12; soma: n = 37) regions according to their β2 subunit densities. Scale bars: (A–D) 200 nm.

use for identifying AISs. Co-localization of the α1 and γ2 (*Figure 11B*) and α2 and γ2 (*Figure 11C*) subunits revealed that both PC somata and AISs are surrounded by relatively weak immunopositive puncta for the α1 subunit and much stronger puncta for the α2 subunit. We noted that despite the very robust demarcation of PC AISs by α2 subunit-immunopositive clusters, all perisomatic synapses were also labeled for the α2 subunit. Antibodies against the β1, β2 and β3 subunits in low-pH fixed tissue visualized strongly immunopositive clusters that delineated the AISs and surrounded the somata of PCs (*Figure 11D–F*). These clusters were extensively co-localized with the γ2 subunit or NL-2, indicating their synaptic nature. Immunofluorescent labeling of the six major GABA$_A$R subunits with a low-pH-based fixation protocol revealed the presence of α1, α2, β1, β2, β3, and γ2 subunits in virtually all AIS and somatic synapses of the CA3 region, consistent with the EM SDS-FRL results. Due to the limited resolution of diffraction-limited confocal microscopy, we refrain from quantitative analysis of our fluorescent reactions.

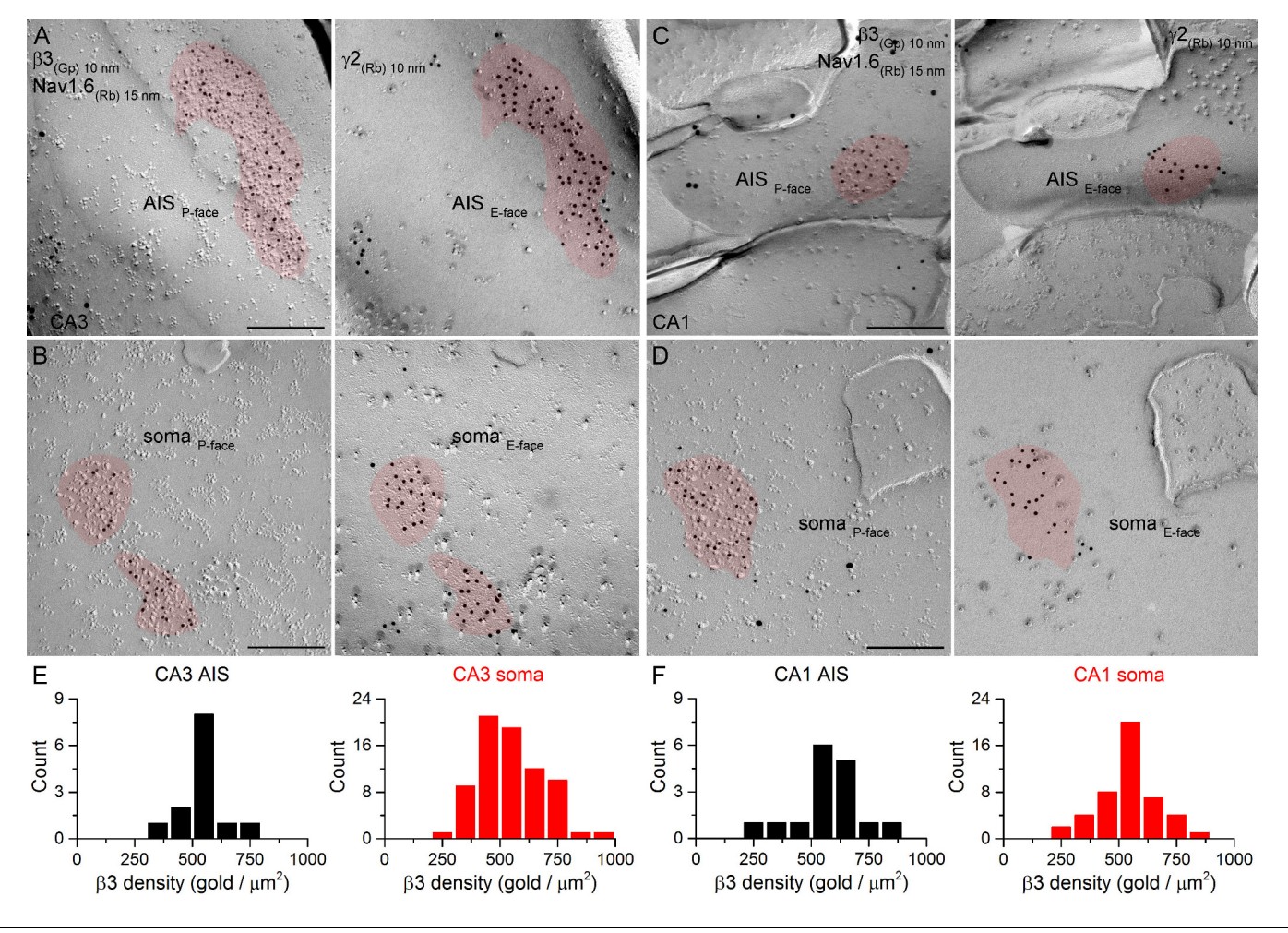

**Figure 10.** SDS-FRL labeling for the β3 subunit in hippocampal PC AIS and somatic synapses. (A–D) High magnification images of double-replica pairs demonstrate the clustering of β3 subunit (10 nm gold) in AIS (A, C) and somatic (B, D) synapses, identified on the complementary E-face by the presence of gold particles labeling the γ2 subunit (10 nm gold). The AIS was identified by immunolabeling for the Nav1.6 subunit (15 nm gold). (E, F) Distributions of AIS (black) and somatic (red) synapses in the CA3 (E; AIS: n = 13; soma: n = 74, one rat) and CA1 (F; AIS: n = 16; soma: n = 46) regions based on their β3 subunit densities. Scale bars: (A–D) 200 nm.

## Discussion

In the present study we investigated the relative abundance of distinct GABA_AR α, β and γ subunit isoforms in hippocampal PC AIS and somatic synapses using the face-matched mirror replica labeling technique. Our results demonstrate that GABA_ARs fracture to both P- and E-face membrane halves, allowing the quantitative comparison of immunogold signals obtained with a single antibody that labels on either side of the membrane. We found that all CA1 PC somatic and AIS synapses contain the α1, α2, β1, β2, β3 and γ2 subunits along with NL-2. More than 90% of CA3 PC perisomatic synapses are immunopositive for the α1 subunit, and all somatic and AIS synapses are positive for all other subunits examined. Somatic synapses formed unimodal distributions based on their GABA_AR subunit and NL-2 content, suggesting the lack of two distinct somatic synapse populations. Finally, we have provided direct evidence for similar α2 subunit densities in synaptic membrane fragments facing Cav2.1[+] (PV) or Cav2.2[+] (CCK) AZs in PC somata.

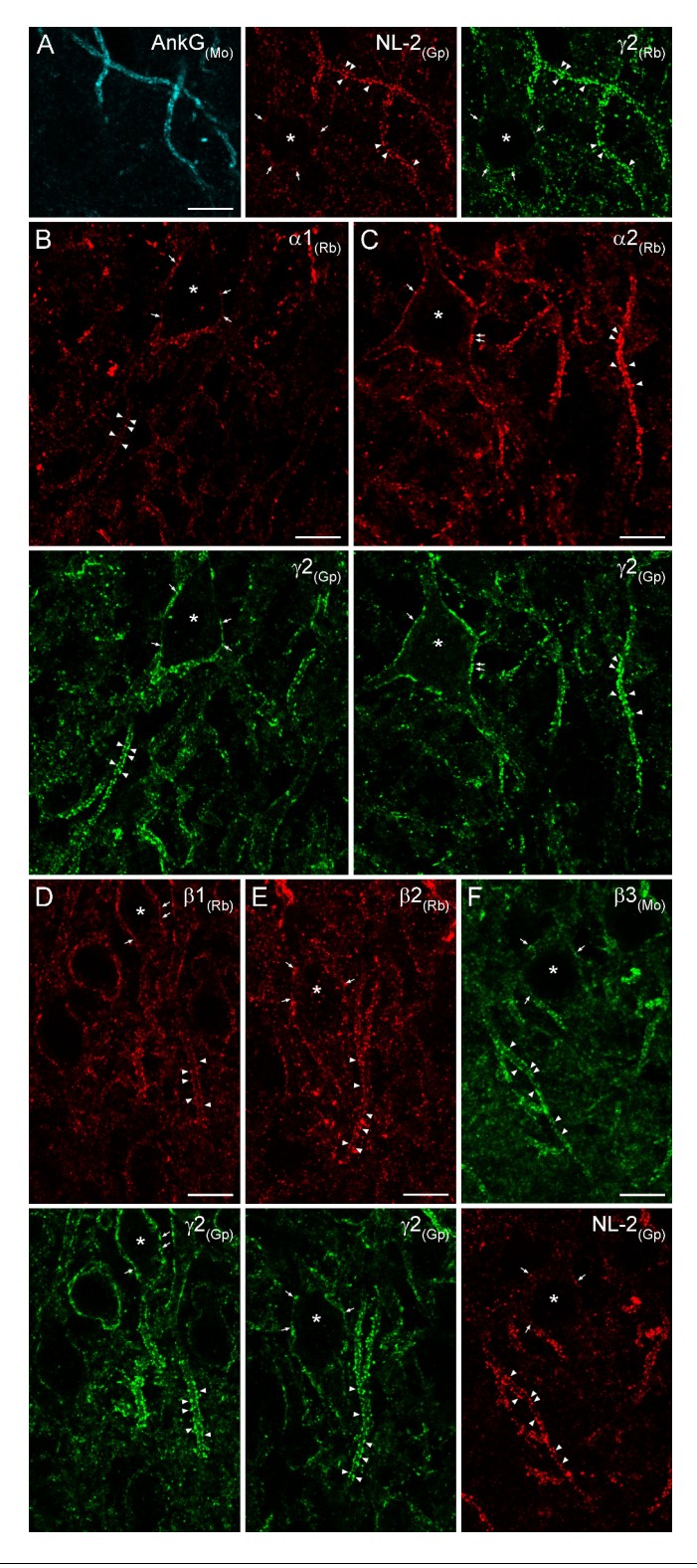

**Figure 11.** Immunofluorescent labeling for six GABA$_A$R subunit isoforms in CA3 PC AIS and somatic synapses. (**A**) Triple-labeling for the AIS marker ankyrin-G (AnkG; blue), the inhibitory synapse marker NL-2 (red) and the γ2 subunit (green) in PC AIS and somatic synapses. Light microscopic image depicts the co-localization of the γ2 subunit with NL-2 clusters in AISs (arrowheads) and somata (*; arrows). (**B–F**) Double-immunofluorescent labeling

*Figure 11 continued on next page*

*Figure 11 continued*

for the α1, α2, β1, β2 (red), β3 (green) and the γ2 subunit (green) or NL-2 (red) reveals the clustering of these GABA$_A$Rs subunit isoforms in AIS (arrowheads) and somatic synapses (*; arrows). (**A–F**) 10 μm.

## Similar GABA$_A$R subunit composition of AIS and somatic synapses of hippocampal PCs

Our quantitative SDS-FRL experiments revealed that the subunit composition of GABA$_A$Rs in somatic and AIS synapses is very similar in both CA1 and CA3 PCs. In agreement with our ultrastructural data, LM immunofluorescent localization of the six GABA$_A$R subunits in low-pH-fixed tissue also revealed that all examined subunits form clusters around PC somata and delineate the AISs. This is consistent with the results of *Panzanelli et al. (2011)* showing the presence of α1 and α2 subunits in both AIS and somatic synapses. However, the presence of every examined subunit in both somatic and AIS synapses does not mean that all synapses contain these subunits at the same density. The relative abundance of a synaptic receptor cannot be easily determined using diffraction-limited LM, and therefore we refrained from quantitative analysis of our fluorescent reactions. In contrast, the SDS-FRL allows the direct quantitative comparison of surface densities between different subcellular compartments. Our quantitative results revealed very similar densities of NL-2 and α1 subunit in somatic and AIS synapses of both CA3 and CA1 regions, but higher α2 and γ2 subunit densities in AIS compared to somatic synapses of CA1 PCs.

Why did several studies conclude that AIS synapses primarily harbor the α2 subunit (*Nusser et al., 1996*; *Fritschy et al., 1998*; *Loup et al., 1998*; *Cruz et al., 2003*)? Prominent demarcation of AISs with α2 subunit-immunopositive clusters is found in many brain regions, including the CA3 area of the hippocampus and in layer 2/3 and 5 PCs of several cortical areas. The demarcation of the AISs in a fluorescent reaction is the consequence of the density of GABAergic synapses on AISs; the larger the synapse density is, the more delineated the AISs will appear. In addition, the strength of a fluorescent synaptic cluster is determined by the density of the receptors and the size of the synapse. The fact that CA3 AISs appear strongly demarcated for the α2 subunit, which has similar densities in somatic and AIS synapses, indicates that the larger synaptic size and the large synaptic coverage of AISs are primarily responsible for their apparent predominance for α2 subunit labeling. Our results presented here also show that the prominent demarcation of CA3 PC AISs is not a unique feature of the α2 subunit, but is also evident for NL-2, β1, β3 and γ2 subunits.

The similar postsynaptic GABA$_A$R subunits in AIS and somatic synapses indicate that AAC- and basket cell-evoked IPSCs should have similar decay kinetics. *Szabo et al. (2010)* compared the decay times of IPSCs evoked by PV$^+$ and CCK$^+$ basket cells and AACs and found that the AAC-evoked postsynaptic response had a slower decay. However, based on the work of *Overstreet and Westbrook (2003)*, demonstrating extensive spillover between the axon terminals of AACs that slows IPSCs, *Szabo et al. (2010)* examined the decay of AAC-evoked IPSCs under low release probability conditions where no spillover is expected and found similar decay times of AAC- and basket cell-evoked IPSCs. All of these results are consistent with the notion that AAC and basket cell activate GABA$_A$Rs with similar subunit composition on their postsynaptic PCs, predicting similar postsynaptic regulation and pharmacological properties.

## Similar α2 subunit densities in somatic synapses established by PV$^+$ or CCK$^+$ basket cells

The presynaptic partners of GABAergic synapses on PC somata are either PV$^+$ or CCK$^+$ basket cell axon terminals. The proposed segregation of α1 and α2 subunit-containing GABA$_A$Rs to synapses established by PV$^+$ and CCK$^+$ basket cells, respectively, comprises one of the most remarkable molecular specializations of synapses that are within the same subcellular compartment separated by only a few microns (*Nyiri et al., 2001*; *Klausberger et al., 2002*). Such a segregation would imply different decay kinetics to the IPSCs, which were not seen using paired recordings between identified basket cells and CA3 PCs (*Szabo et al., 2010*). Using transgenic animals with a point mutation rendering the α2 subunit insensitive to benzodiazepines, *Heistek et al. (2013)* demonstrated that fast-spiking (PV$^+$) basket cells activate α2 subunit-containing GABA$_A$Rs in CA3 PCs. These results,

taken together with our quantitative EM data showing similar α2 subunit densities in the two basket cell synapse populations, point towards a similar subunit composition of postsynaptic GABA$_A$Rs in these functionally distinct somatic synapse populations.

## Variability in the ratio of GABA$_A$R α subunit isoforms in different neurons

Most central neurons express multiple GABA$_A$R subunit types (*Fritschy and Mohler, 1995*; *Pirker et al., 2000*) and their individual synapses contain many different subunits. A study from our laboratory demonstrated that external tufted cells of the main olfactory bulb express the α1 and α3 subunits and display small synapse-to-synapse, but large cell-to-cell variability in the α1:α3 ratio (*Eyre et al., 2012*). Consistent with this, the decay kinetics of IPSCs recorded from tufted cells have small within-cell, but large cell-to-cell variability. Here, we found that hippocampal PCs exhibit moderate synapse-to-synapse, but small cell-to-cell variability in the α1:α2. This is in agreement with the very small cell-to-cell variability in mIPSC decay times recorded from PCs. Thus, our analysis of the ratios of two α subunits in two cell types demonstrate that distinct neuronal types use unique strategies for expressing multiple synaptic GABA$_A$R α subunit isoforms with either a variable or constant ratio from synapse-to-synapse and cell-to-cell, allowing them to fulfill individual cellular requirements in network dynamics.

## Technical considerations

The discrepancy between results acquired with postembedding immunogold localizations (*Nusser et al., 1996*; *Nyiri et al., 2001*; *Klausberger et al., 2002*) and those obtained with SDS-FRL (this study and *Kasugai et al., 2010*) might lie in the difference in the sensitivity of the two methods. The highest labeling efficiency of the postembedding immunogold method for AMPA and GABA$_A$Rs localization was estimated to be around 30–40% (*Nusser et al., 1997*, *1998*), but it could be as high as 90–100% with SDS-FRL (*Tanaka et al., 2005*; *Lorincz and Nusser, 2010*). The density of functional GABA$_A$Rs was estimated to be ~1200 receptors per μm$^2$ in cerebellar stellate cell synapses (*Nusser et al., 1997*). Our face-matched mirror replica labeling for the γ2 subunit revealed around 800 gold / μm$^2$ on the E- and 250 gold / μm$^2$ on the P-face, resulting in ~1050 gold in every μm$^2$ of postsynaptic membrane. If a similar GABA$_A$R density characterizes hippocampal perisomatic synapses to that found in cerebellar stellate cells, an almost 90% labeling efficiency can be calculated in our best reactions. Thus, SDS-FRL has a superior sensitivity compared to postembedding labeling, the only other localization technique that allows the visualization of receptors and ion channels embedded in dense protein matrices found in e.g. postsynaptic membranes or AISs. Increasing the sensitivity of a localization method results in the disappearance of negative synapses if their negativity was the consequence of a low protein content and not the lack of the protein. It is important to emphasize that even with SDS-FRL, some of the synapses could remain immunonegative (e.g. some of the CA3 PC perisomatic synapses were apparently negative for the α1 subunit) and it is erroneous to conclude that their negativity means the lack of the examined protein just because this technique with certain antibodies yields a labeling efficiency close to 100%. The fact that >99% of synapses are immunopositive in CA1 PCs, where the overall density of the α1 subunit is >2 fold larger than in CA3 PCs, indicates that the 10% immunonegativity of the CA3 synapses is probably due to low subunit densities and not the lack of the subunit. Concluding the lack of a protein in a subcellular compartment with any immunolocalization technique requires a labeling efficiency close to 100% and this needs to be determined for each antibody.

## Materials and methods

All experiments were conducted in accordance with the Hungarian Act of Animal Care and Experimentation (1998, XXVIII, section 243/1998) and with the ethical guidelines of the Institute of Experimental Medicine Protection of Research Subjects Committee.

### Fluorescent immunohistochemistry

Three adult male Wistar rats were anaesthetized with isoflurane and ketamine, then transcardially perfused with a fixative containing 1% paraformaldehyde (PFA) in 0.1 M Na-acetate buffer for 13 min. 70 μm coronal section were cut with a Vibratome and were washed in phosphate buffer (PB).

After several washes in Tris-buffered saline (TBS), the sections were blocked with 10% normal goat serum (NGS) for 1 hr. Then, sections were incubated in a solution containing a mixture of primary antibodies made up in TBS containing 0.1% Triton-X and 2% NGS overnight at 4°C. The primary antibodies used for fluorescent immunohistochemistry are listed in *Table 1*. Next, sections were incubated in Alexa488-conjugated goat anti-rabbit IgGs (1:500, Life Technologies), Daylight488-conjugated donkey anti-guinea-pig IgGs (1:500, Jackson ImmunoResearch), Cy3-conjugated goat anti-rabbit or donkey anti-guinea-pig IgGs (1:1000, Jackson ImmunoResearch) and Cy5-conjugated goat anti-mouse IgGs (1:1000, Jackson ImmunoResearch) secondary antibodies made up in TBS containing 2% NGS for 2 hr. Images were acquired using a confocal laser scanning microscope (FW100 Olympus) with a 60X oil-immersion objective (numerical aperture (NA) = 1.35).

## Sodium dodecyl sulphate-digested freeze-fracture replica labeling

Eleven adult male Wistar rats were anaesthetized with isoflurane and ketamine, then transcardially perfused with a fixative containing 2% PFA and ~0.2% picric acid in 0.1 M PB for 16 min. 80 µm coronal sections were cut and small tissue blocks of the CA1 and CA3 areas were cut out from the hippocampus. The blocks were frozen with a high-pressure freezing machine (HPM100, Leica Microsystems), fractured with a freeze-fracture machine (BAF060, Leica) and processed for SDS-FRL as described previously (*Kerti-Szigeti et al., 2014*). Briefly, the replicas were treated with TBS containing 2.5% SDS at 80°C for 18 hr. Replicas were then washed in TBS, followed by blocking with 0.1–5% bovine serum albumin (BSA) for 1 hr. Next, replicas were incubated in blocking solution containing a mixture of primary antibodies. The list of primary antibodies applied for SDS-FRL is

**Table 1.** List of primary antibodies used in fluorescent immunohistochemistry and SDS-FRL.

| Molecule | Host | Epitope (aa residues) | Vendor | RRID/Cat. No. | Protein concentration | Fluorescent reaction (dilution) | SDS-FRL (dilution) |
|---|---|---|---|---|---|---|---|
| GABA$_A$R α1 | Rb | 1–9 | W. Sieghart | | 958 µg/ml | 1:1000 | 1:200 |
| GABA$_A$R α1 | Mo | 28–43 | Synaptic Systems | 224 211 | 1 mg/ml | | 1:60 |
| GABA$_A$R α2 | Rb | 29–37 | Synaptic Systems | 224 103 | 1 mg/ml | 1:1000 | |
| GABA$_A$R α2 | Rb | 322–357 | W. Sieghart | | 782/991 µg/ml | | 1:10000/1:700 |
| GABA$_A$R α2 | Gp | 1–9 | J.-M. Fritschy | AB_2314463 | | | 1:100 |
| GABA$_A$R β1 | Rb | 342–430 | Synaptic Systems | 224 703 | 1 mg/ml | 1:1000 | 1:1000 |
| GABA$_A$R β1 | Gp | 342–430 | Synaptic Systems | 224 705 | 1 mg/ml | | 1:1200 |
| GABA$_A$R β2 | Rb | 343–430 | Synaptic Systems | 224 803 | 1 mg/ml | 1:1000 | |
| GABA$_A$R β2 | Gp | 343–430 | Synaptic Systems | 224 805 | 1 mg/ml | | 1:600 |
| GABA$_A$R β3 | Rb | 345–408 | W. Sieghart | | 479 µg/ml | | 1:800 |
| GABA$_A$R β3 | Gp | 344–429 | Synaptic Systems | 224 404 | serum | | 1:500 |
| GABA$_A$R β3 | Mo | 370–433 | Neuromab | AB_2109585 | | 1:1000 | |
| GABA$_A$R γ2 | Rb | 39–67 | Synaptic Systems | 224 003 | 1 mg/ml | 1:1000 | 1:600 |
| GABA$_A$R γ2 | Rb | 319–366 | W. Sieghart | | 270 µg/ml | | 1:1000 |
| GABA$_A$R γ2 | Gp | 39–67 | Synaptic Systems | 224 004 | 1 mg/ml | 1:500 | |
| GABA$_A$R γ2 | Mo | 39–67 | Synaptic Systems | 224 011 | 1 mg/ml | | 1:600 |
| neuroligin-2 | Gp | 732–761 | Frontier Institute | AB_2571609 | 200 µg/ml | 1:100 | 1:50 |
| neuroligin-2 | Gp | 750–767 | Synaptic Systems | 129 205 | 1 mg/ml | | 1:600 |
| ankyrin-G | Mo | 990–2622 | Neuromab | AB_10673030 | 1 mg/ml | 1:500 | |
| Nav1.6 | Rb | 1042–1061 | Alomone | AB_2040202 | 0.8 mg/ml | | 1:600 |
| Kv1.1 | Gp | 478–492 | Frontier Institute | Kv1.1-GP-Af1000 | 200 µg/ml | | 1:50/1:100 |
| Cav2.1 | Gp | 1921–2212 | Synaptic Systems | 152 205 | 1 mg/ml | | 1:1000 |
| Cav2.2 | Rb | 2056–2336 | Synaptic Systems | 152 303 | 1 mg/ml | | 1:1000 |

Abbreviations: aa, amino acids; Gp, guinea pig; Mo, mouse; Rb, rabbit.

provided in *Table 1*. For the majority of single- or double-immunogold labeling, replicas were incubated in the primary antibody solution for one night at room temperature, except for GABA$_A$R α1 subunit labeling, where replicas were incubated for four days at 4°C. The following day, replicas were incubated in blocking solution containing the following secondary antibodies: 10 nm gold-conjugated goat anti-rabbit or goat anti-guinea-pig IgGs, or 15 nm gold-conjugated goat anti-rabbit or goat anti-guinea-pig IgGs (1:100 applies to all; British BioCell International (BBI)). For some double-replica labeling the antibodies were applied sequentially as follows. On the first day, a guinea-pig anti-Kv1.1 was applied, followed by 15 nm gold-conjugated goat anti-guinea-pig IgG (1:100, BBI). This was followed by the application of the second primary antibody (rabbit anti-GABA$_A$R α2) and then the corresponding second secondary antibody (10 nm gold-conjugated goat anti-rabbit IgG, 1:100, BBI). For triple-replica labeling the antibodies were applied in the following order. First, replicas were double-labeled for guinea-pig anti Cav2.1 and rabbit anti-Cav2.2 for one night at room temperature. Next, 15 nm gold-conjugated goat anti-guinea-pig IgG (1:50, BBI) and 10 nm gold-conjugated goat anti-rabbit IgG (1:100, BBI) were applied to label the primary antibodies, followed by the application of the third primary antibody (mouse anti-GABA$_A$R γ2) overnight at room temperature. Finally, 5 nm gold-conjugated goat anti-mouse IgG (1:100, BBI) was applied to label the third primary antibody. After the immunoreactions, replicas were rinsed in distilled water and were picked up on copper grids and examined with a transmission electron microscope (TEM, JEM1011, Jeol).

Antibodies used in this study recognized either intracellular or extracellular epitopes on their target proteins and consequently were visualized by gold particles labeling on the protoplasmic face (P-face) or the exoplasmic face (E-face), respectively. The nonspecific background labeling was measured on either E-face or P-face structures of the replicas, depending on the specific labeling of the target proteins. Background labeling was very low for NL-2 and the GABA$_A$R subunits, and therefore was not subtracted from the synaptic density values.

## Quantification of immunogold labeling in AIS and somatic synapses

To identify the AIS of hippocampal PCs, immunogold labeling for Nav1.6 or Kv1.1 subunits was carried out (*Lorincz and Nusser, 2010*; *Kirizs et al., 2014*). A GABAergic synapse-specific adhesion molecule (*Varoqueaux et al., 2004*), NL-2, was used to identify AIS and somatic synapses on the P-face of the plasma membrane, where GABAergic synapses appear as small membrane patches rich in IMPs (*Kasugai et al., 2010*). To quantitatively analyze the density of gold particles labeling different GABA$_A$R subunits in perisomatic synapses, the face-matched mirror replica technique was used (*Hagiwara et al., 2005*; *Kerti-Szigeti et al., 2014*). Replicas were generated from both faces of the fractured tissue surfaces, allowing the simultaneous labeling of the complementary E- and P-faces of the same plasma membranes. Synaptic areas were delineated on the P-face based on immunogold labeling for NL-2 or the α2, β1–3 and γ2 subunits of the GABA$_A$Rs and on the IMP clusters. The complementary E-face image was then superimposed on the P-face image and the synaptic area was projected onto the E-face image. Gold particles inside the synaptic area and up to 30 nm away from its edge (*Matsubara et al., 1996*) were counted on both faces. To identify synapses established by PV$^+$ or CCK$^+$ basket cells, one face of the replica was triple-labeled for Cav2.1 and Cav2.2 Ca$^{2+}$ channel subunits, as markers for PV$^+$ and CCK$^+$ axon terminals, respectively (*Lenkey et al., 2015*), and the γ2 subunit to label the GABAergic synapses. The complementary face was labeled for the α2 subunit. The α2 content of the two synapse populations was assessed in partially fractured postsynaptic membranes facing Cav2.1-positive (Cav2.1$^+$, i.e. PV$^+$) or Cav2.2-positive (Cav2.2$^+$·i.e. CCK$^+$) active zones (AZs). All other GABA$_A$R density measurements were performed on synapses fractured in their entirety.

## Data analysis and statistical tests

Quantification of immunogold labeling for NL-2 and the GABA$_A$R α1, α2 and γ2 subunits was repeated in two to five rats (the exact number of rats used for quantification is indicated in the results and figure legends), while that for the β1, β2 and β3 subunits was performed in one rat. The reaction intensities varied between individual animals, probably due to differences in fixation. Therefore, in order to pool the synaptic density values for statistical comparisons, each density value was normalized to the reaction average, which was calculated by averaging density values for AIS and somatic synapses in both CA3 and CA1 areas. Normalized density values for NL-2, γ2, α1 and α2

subunits are shown in *Figures 2H,I*, *3H–K*, *4G–I*, *5G–I*. Raw density values are shown whenever quantification was made only in one rat (immunogold labeling for β subunits), or when the reactions were not different between rats, and hence they could be pooled for statistical comparisons (i.e. α2 density measurement in identified synapses, *Figure 7*; p>0.0125, after Bonferroni correction following Mann-Whitney tests for pair-wise comparison of α2 density values in four compartments for two rats: identified (Cav2.1$^+$ and Cav2.2$^+$) and unidentified synapses, and E-face background labeling). The α1 versus α2 density ratios (α1:α2) were calculated by dividing normalized α1 and α2 density values of individual synapses (*Figure 6E*).

All statistical comparisons were made with Statistica11 software (Scientific Computing). Normality of sample distribution was assessed using the Shapiro-Wilk test. For correlation analyses the Spearman's rank order correlation was used as most of the samples were not normally distributed. Statistical significance was assessed with the Mann-Whitney U test with Bonferroni correction, or the Kruskal-Wallis test followed by *post hoc* multiple comparisons of mean ranks for all groups. Significance was taken at p<0.05 (*), p<0.01 (**) or p<0.001 (***). Medians and lower ($Q_1$) and upper quartiles ($Q_3$) were used to describe distributions throughout the manuscript.

## Specificity of the immunoreactions

Specificity of immunogold labeling for the α1, α2 and γ2 subunits was verified by using two antibodies directed against different epitopes of the same protein. Similar labeling patterns were obtained with a rabbit anti-α1 (α1$_{(Rb;\ aa1-9)}$) and a mouse anti-α1 (α1$_{(Mo;\ aa28-43)}$) antibody, indicating the specificity of the reaction. Mirror replica labeling was used to assess the specificity of our α2 and γ2 labeling, by using antibodies directed against an extracellular and an intracellular epitope. We observe similar gold particle labeling patterns with a rabbit anti-α2 antibody (α2$_{(Rb;\ aa322-357)}$) on the P-face compared to that obtained with a guinea-pig anti-α2 antibody (α2$_{(Gp;\ aa1-9)}$) on the E-face. Our γ2 labeling on the E-face, obtained with a rabbit antibody (γ2$_{(Rb,\ aa39-67)}$), was very similar to the immunogold labeling seen on the P-face with a rabbit anti-γ2 antibody (γ2$_{(Rb,\ aa319-366)}$), recognizing an intracellular epitope (*Figure 1A–D*). We could not purchase anti-β subunit antibodies raised against different epitopes, and therefore we could not test the specificity of the labeling using two antibodies. However, we performed SDS-FRL immunogold labeling for the β1 and β2 subunits in brain areas and nerve cells where the genes of these subunits are not expressed (e.g. cerebellar Purkinje cells, and medial habenula neurons), and observe very few gold particles labeling for β1 or β2 subunits in GABAergic synapses (zero or 1–3 gold particles / synapse; data not shown). These results indicate that our immunogold labeling for the β1 (β1$_{(Gp;\ aa342-430)}$) and β2 (β2$_{(Gp;\ aa343-430)}$) subunits in hippocampal perisomatic synapses is probably due to specific antibody-protein interactions. We observed a similar labeling pattern with our guinea-pig anti-β3 antibody (β3$_{(Gp;\ aa344-429)}$) to that published by *Kasugai et al. (2010)* with a different antibody against the β3 subunit, the specificity of which was proven in β3$^{-/-}$ mice.

## Acknowledgements

ZN is the recipient of a Hungarian Academy of Sciences Momentum Grant (Lendület, LP2012-29) and a European Research Council Advanced Grant (293681). The financial support from these funding bodies is gratefully acknowledged. We would like to thank Drs. Peter Somogyi, Thomas Klausberger, Gabor Nyiri and Mark Eyre for their comments on the manuscript; Drs. Jean-Mark Fritschy and Werner Sieghart for kindly providing GABA$_A$R-specific antibodies. We thank Éva Dobai for her excellent technical assistance.

## Additional information

### Funding

| Funder | Grant reference number | Author |
| --- | --- | --- |
| European Research Council | 293681 | Zoltan Nusser |
| Magyar Tudományos Akadémia | LP2012-29 | Zoltan Nusser |

The funders had no role in study design, data collection and interpretation, or the decision to submit the work for publication.

## Author contributions

KK-S, Conception and design, Acquisition of data, Analysis and interpretation of data, Drafting or revising the article; ZN, Conception and design, Analysis and interpretation of data, Drafting or revising the article

## Author ORCIDs

Katalin Kerti-Szigeti, http://orcid.org/0000-0001-9500-8758
Zoltan Nusser, http://orcid.org/0000-0001-7004-4111

## Ethics

Animal experimentation: All experiments were conducted in accordance with the Hungarian Act of Animal Care and Experimentation (1998, XXVIII, section 243/1998) and with the ethical guidelines of the Institute of Experimental Medicine Protection of Research Subjects Committee.

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
