## [Decision Letter]

Thank you for submitting your article "Similar GABA_A_ receptor subunit composition in somatic and axon initial segment synapses of hippocampal pyramidal cells" for consideration by *eLife*. Your article has been reviewed by three peer reviewers, one of whom, Marlene Bartos, is a member of our Board or reviewing Editors, and another is Huibert D. Mansvelder (Reviewer #2). The evaluation has been overseen by Gary Westbrook as the Senior Editor. The reviewers have discussed the reviews with one another and the Reviewing Editor has drafted this decision to help you prepare a revised submission.

Summary:

The Reviewing Editor and two reviewers considered your work as a thorough and high quality quantification of GABA_A_ receptor subunits expressed at synapses on soma and axon initial segments of hippocampal pyramidal cells, which provides a decisive answer to the apparent controversy reported in previous studies. The data were judged as highly convincing and the conclusions are supported by the data. However, the reviewers raised several concerns that require your attention.

Essential revisions:

1) Figure 1: from the images displayed it appears there are many more particles (IMP) in the p-face than in the e-face images, which is particularly apparent in C and D. What the nature of these particles is, is not clear, but a subset of them are GABA_A_Rs. Given the unequal distribution of particles between p-face and e-face, GABA_A_Rs could hypothetically also be unequally distributed between the two faces. If the two antibodies used to recognize the intracellular and the extracellular epitopes have different efficacy in labeling GABA-R γ2 subunits, e.g. different affinities, this could results in a unequal number of gold particles visible in both images. This would jeopardize the validity of the conclusion that GABA_A_Rs equally end up in the p-face and e-face. What evidence do the authors have to show that the two antibodies have similar labeling efficacy?

2) The data presented in Figure 4 shows that CA3 synapses are more often devoid of α1 subunits, whereas CΑ1 synapses always contain α1 subunits. Are these α2 dominated synapses without α1 subunits on CA3 pyramidal cells preferentially in synpases made by PV+ interneurons, as is suggested in the literature (e.g. Heistek et al., 2013)? Why was the synapse-type analysis of Figure 7 using Cav2.1+ (PV) or Cav2.2+ (CCK) only done in CA1 and not in CA3?

3) Figure 6: From the distributions in panel E it seems there could be a difference in distribution of the α1:α2 ratios between CA1 and CA3 synapses. From the text on page 12 describing the results of these data, it was not clear what exactly was compared. It reads "However, our analysis revealed a lack of significant difference between the α1:α2 in AIS and somatic synapses in the CA1 (p = 0.347, Kruskal-Wallis test, followed by multiple comparisons of mean ranks) and CA3 (p = 0.053) regions (Figure 6)". This suggests that soma and AIS were compared in the areas separately. Were the α1:α2 ratio distributions statistically different between CA1 and CA3 synapses (similar to the analysis done for Figure 4)?

4) Figure 11: there is a strong difference in the pattern of α1 and α2 subunits expression in panel C and D. The conclusion from these light microscopy data could be that the EM data may not resemble the light microscopy data, in particular for the α1 and α2 subunits. The lack of quantification of these data does not facilitate interpretation. Perhaps if similar immunostaining at the light microscopy level for CA1 were shown as well, they would be more convincing and show a similar distribution for α1 and α2 subunits.

5) How confident are the authors that all immunogold-labeled GABA_A_Rs subunits are part of functional receptors? This should be discussed.

6) The abstract does not reflect all major findings. It mentions the results on the CA1 synapses, but no mention of the CA3 data and comparisons between the areas.

---

## [Author Response]

*Essential revisions:*

*1) Figure 1: from the images displayed it appears there are many more particles (IMP) in the p-face than in the e-face images, which is particularly apparent in C and D. What the nature of these particles is, is not clear, but a subset of them are GABA_A_Rs. Given the unequal distribution of particles between p-face and e-face, GABA_A_Rs could hypothetically also be unequally distributed between the two faces. If the two antibodies used to recognize the intracellular and the extracellular epitopes have different efficacy in labeling GABA-R γ2 subunits, e.g. different affinities, this could results in a unequal number of gold particles visible in both images. This would jeopardize the validity of the conclusion that GABA_A_Rs equally end up in the p-face and e-face. What evidence do the authors have to show that the two antibodies have similar labeling efficacy?*

We have never claimed that GABA_A_Rs *equally* fracture to the P- and E-faces, instead we provided experimental evidence showing that GABA_A_Rs fracture to both sides of the plasma membrane. The variance around an imaginative regression line in Figure 1 is likely to be the consequence of some randomness in fracturing.

In Figure 1 we show that there are twice as many gold particles in the E- compared to the P-face of the synapses. This could be the consequence of a similar labeling efficacy and twice as many GABA_A_Rs on the E-face of the synapses, or similar number of GABA_A_Rs on both faces with a higher labeling efficacy with our antibody that labels the E-face, or the combination of both. There is no way we can distinguish between these possibilities. It would be interesting to know this, but we believe that this is not the most important point. The most important point is the positive correlation between the signals obtained on E- and P-faces, demonstrating that if there are lots of gold particles on one side, then there will be many gold particles on the other side as well.

*2) The data presented in Figure 4 shows that CA3 synapses are more often devoid of α1 subunits, whereas CA1 synapses always contain α1 subunits. Are these α2 dominated synapses without α1 subunits on CA3 pyramidal cells preferentially in synpases made by PV+ interneurons, as is suggested in the literature (e.g. Heistek et al., 2013)? Why was the synapse-type analysis of Figure 7 using Cav2.1+ (PV) or Cav2.2+ (CCK) only done in CA1 and not in CA3?*

We believe that the 10% immunonegativity of CA3 PC perisomatic synapses is unlikely the consequence of multiple synapse populations, but more likely due to low α1 densities that remain below our detection threshold in the weakest synapses. From these experiments, it is impossible to tell whether these low α1-containing synapses face PV^+^ or CCK^+^ axon terminals. In our original version, we restricted our detailed analysis of the α2 subunit in Cav2.1- or Cav2.2-positive synapses to the CA1 area because the previous studies were all performed in this subregion (Nyiri et al., 2001; Klausberger et al., 2002) and because collecting such synapses is extremely challenging. Based on the suggestion of the reviewers, we have performed the same analysis now in the CA3 area and included our data in Figure 7. The results in the CA3 area are consistent with our data obtained in the CA1 area. We have described our new results in the Results section, subsection “Somatic synapses established by PV+ and CCK^+^ axon terminals have similar α2 subunit densities”.

*3) Figure 6: From the distributions in panel E it seems there could be a difference in distribution of the α1:α2 ratios between CA1 and CA3 synapses. From the text on page 12 describing the results of these data, it was not clear what exactly was compared. It reads "However, our analysis revealed a lack of significant difference between the α1:α2 in AIS and somatic synapses in the CA1 (p = 0.347, Kruskal-Wallis test, followed by multiple comparisons of mean ranks) and CA3 (p = 0.053) regions (Figure 6)". This suggests that soma and AIS were compared in the areas separately. Were the α1:α2 ratio distributions statistically different between CA1 and CA3 synapses (similar to the analysis done for Figure 5 and Figure 4)?*

We regret that description of our results was not clear. The α1:α2 density ratios in AIS and somatic synapses in both CA1 and CA3 regions were compared using the non-parametric Kruskall-Wallis test (p < 0.001), followed by multiple comparisons of mean ranks *post-hoc* test of the four compartments. The α1:α2 ratio was significantly lower in CA3 AIS compared to CA1 somatic synapses (p = 0.026), and in CA3 somatic synapses compared to CA1 AIS (p = 0.001) and CA1 somatic synapses (p < 0.001). But no significant difference was found in the α1:α2 ratios in AIS vs. somatic synapses in both CA1 (p = 0.347) and CA3 regions (p = 0.053).

In this paragraph we aimed at providing a comparison of somatic vs. AIS synapses, therefore we refrained from the detailed description of the statistical results obtained after multiple comparisons between compartments of different hippocampal areas. This would not add additional information to the main argument of the paragraph showing the lack of significant difference in α1:α2 ratios between AIS and somatic synapses (i.e. similar α1:α2 variability).

*4) Figure 11: there is a strong difference in the pattern of α1 and α2 subunits expression in panel C and D. The conclusion from these light microscopy data could be that the EM data may not resemble the light microscopy data, in particular for the α1 and α2 subunits. The lack of quantification of these data does not facilitate interpretation. Perhaps if similar immunostaining at the light microscopy level for CA1 were shown as well, they would be more convincing and show a similar distribution for α1 and α2 subunits.*

The reviewers are correct in pointing out that the results of the LM immunofluorescent reactions are difficult to compare to the quantitative data obtained with replica labeling. However, we believe that our LM data is qualitatively consistent with the EM quantification. For example, AIS synapses are all immunopositive for the α1 subunit, although the intensity of the clusters are substantially weaker compared to those labeled for the α2, β1, β3 or γ2 subunits. This is also true for most somatic synapses. There are many reasons why we refrain from performing quantification of the LM reactions. One of the main reasons is the variability in the results depending on the fine details of the tissue preparation (e.g. antigen retrieval, the used fixative).

*5) How confident are the authors that all immunogold-labeled GABA_A_Rs subunits are part of functional receptors? This should be discussed.*

This is a fundamental question that generally applies to all immunolocalization experiments. The possibility can never be excluded that some of the proteins are in an ‘inactive’ state and they contribute to the immunosignal without adding to the function. There are studies (Nusser, Neuron, 1997; Tanaka, J Neurosci, 2005; Lorincz, Science, 2010), however, showing correlations between the number of channels/receptors as determined with EM immunogold methods and with functional approaches. Such correlations would only occur if either all receptors/channels were active or a fix proportion of them were inactive. In either case, the results of the immunolocalization experiments showing relative differences in receptor densities between distinct synapses would hold predictive power to function.

*6) The abstract does not reflect all major findings. It mentions the results on the CA1 synapses, but no mention of the CA3 data and comparisons between the areas.*

Based on the suggestion of the reviewers, we rephrased the abstract and included the results obtained from the CA3 region.